# A Structured, Tagged, and Localized Visual Question Answering Dataset with Full Sentence Answers and Scene Graphs for Chest X-ray Images

## Abstract

Visual Question Answering (VQA) enables targeted and context-dependent analysis of medical images, such as chest X-rays (CXRs). However, existing VQA datasets for CXRs are typically constrained by simplistic and brief answer formats, lacking localization annotations (e.g., bounding boxes) and little metadata (e.g., region or radiological finding/disease tags). To address these limitations, we introduce *MIMIC-Ext-CXR-QBA* (abbr. *CXR-QBA*), a large-scale CXR VQA dataset derived from MIMIC-CXR, comprising 42 million QA-pairs with multi-granular, multi-part answers, detailed bounding boxes, and structured tags. We automatically generated our VQA dataset from scene graphs (also made available), which we constructed using LLM-based information extraction from radiology reports. After automatic quality assessment, we identified 31M pre-training and 7.5M fine-tuning grade QA-pairs, providing the largest and most sophisticated VQA dataset for CXRs to date. Tools for using our dataset and the construction pipeline are available at `https://anonymous.4open.science/r/mimic-ext-cxr-qba/`.

## 1 Introduction

With the emergence of Large Language Models (LLMs) and Large Multimodal Models (LMMs), interactive and conversational tasks have gained popularity in medical image analysis, particularly in the context of *chest X-ray (CXR)* interpretation [1]–[7]. A prominent example of such interactive tasks is *Visual Question Answering (VQA)*, where a model is presented with an image and a corresponding textual question, and is tasked with generating an answer. Unlike conventional medical imaging approaches, which always produce the same output (such as classification labels, bounding boxes, or textual reports) for a given image, VQA enables users to interactively explore and interpret images in a context-dependent manner. Training robust VQA models for medical applications requires high-quality, large-scale training datasets. Existing CXR VQA datasets [1], [8]–[13] suffer from several limitations: (i) they often contain only short and simplistic answers, (ii) they lack localization information (such as bounding boxes), and (iii) they provide little structured metadata (e.g., region and finding/disease annotations, or uncertainty estimates). Additionally, their relatively small size constrains their utility for pretraining.

To address these challenges, we propose a pipeline for automatic VQA dataset creation and apply it to construct a new large-scale CXR VQA dataset. Unlike prior datasets, each question-answer (QA) pair includes multi-granular, multi-part answers composed of full sentences in the style of radiology reports. Furthermore, our dataset provides detailed bounding boxes and additional structured tags (e.g., findings and regions), enhancing interpretability and facilitating the development of more advanced and transparent medical VQA models. Fig. 1 shows examples of our generated QA-pairs.

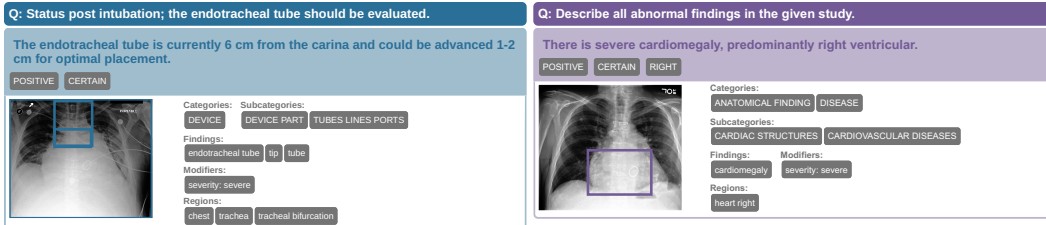

(a) **Indication** question.      (b) **Study abnormality** question.

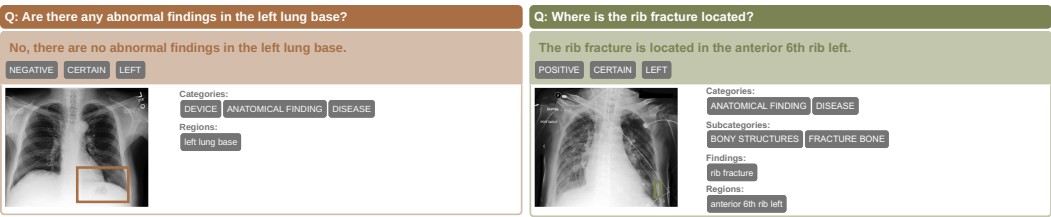

(c) **Region abnormality** question.      (d) **Finding** question.

Figure 1: Examples of question-answer (QA) pairs for each of our four different types of questions. For each question (for a given chest X-ray), a detailed answer with sentences in the style of free-text radiology reports is given, supplemented by bounding boxes (for both positive and negative answers), and a set of tags (e.g. regions, findings, certainty, etc.). For more examples, we refer to Appendix A.

Our contributions are as follows:

- We propose an automatic scene graph construction method as an intermediate step for VQA dataset creation, utilizing LLMs, semantic entity mapping, and localization models.
- We propose a question-answer generation strategy based on the extracted scene graphs.
- Building on this approach, we introduce *MIMIC-Ext-CXR-QBA* (abbr. *CXR-QBA*), a 42M QA-pair VQA dataset derived from MIMIC-CXR [14], to be published on Physionet [15].
- We automatically evaluate the quality of the generated QA-pairs, identifying 31.2M pairs as pre-training grade and 7.5M of these as fine-tuning grade.
- We provide a detailed analysis of our dataset and demonstrate its utility on the newly proposed structured VQA task.

## 2 Related Work

**VQA Datasets for Chest X-Rays** VQA datasets (shown in Tab. 1) are scarce in the medical imaging domain, with most notable examples being VQA-RAD [8] and SLAKE [9], which are hand-labeled but limited in size (3.5K and 14K QA-pairs, respectively). On the other hand, VQA-Med at ImageCLEF 2019 [10] was automatically constructed using QA templates based on image annotations, which may limit its answer quality. To improve the quality, PMC-VQA [11] used an LLM to generated QA-pairs based on provided captions. VQA datasets for chest X-rays include MIMIC-Ext-MIMIC-CXR-VQA [12], [15] and Medical-CXR-VQA [13], [15], [16], which contain hundreds of thousands of QA-pairs (in these cases derived from MIMIC-CXR). These datasets rely on templates but use radiology reports as their original information source, where MIMIC-

Table 1: Comparison of medical VQA datasets. We present the currently largest dataset, additionally providing boxes and tags for the answers.

| Dataset | #QA | Boxes | Tags | Answers |
|---|---|---|---|---|
| **CXR-QBA (Ours)** | **42.2M** | ✓ | ✓ | detailed |
| VQA-RAD [8] | 3.5K | ✗ | ✗ | brief |
| SLAKE [9] | 14K | ✗ | ✗ | brief |
| ImageCLEF [10] | 15K | ✗ | ✗ | brief |
| PMC-VQA [11] | 227K | ✗ | ✗ | brief |
| MIMIC-CXR-VQA [12] | 377K | ✗ | ✗ | brief |
| Medical-CXR-VQA [13] | 780K | ✗ | ✗ | brief |
| CheXinstruct [1] | 8.5M | ✗ | ✗ | brief |

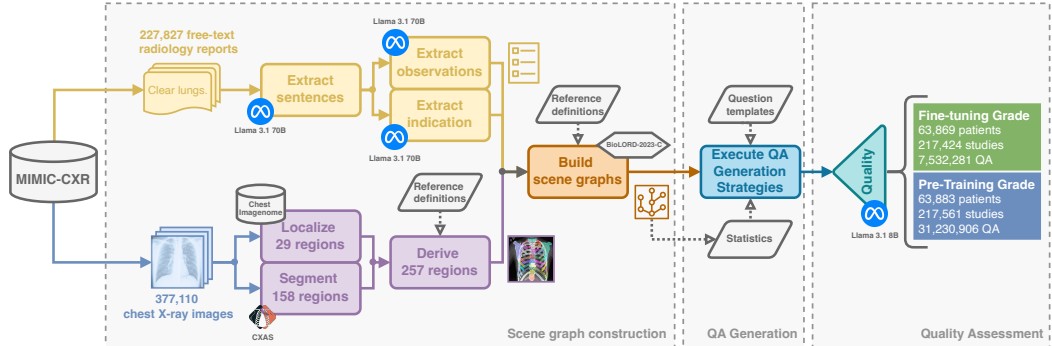

Figure 2: Overview of our dataset construction pipeline. First, we construct scene graphs based on information extracted from the radiology reports and regions localized in the images. Next, we generate question-answer pairs based on templates and the scene graphs. Finally, we automatically assess the quality of generated QA-pairs.

Ext-MIMIC-CXR-VQA leverages Chest ImaGenome's [17] scene graphs and Medical-CXR-VQA employs an LLM-based extraction strategy similar to ours but without semantic entity mapping, localization, and extraction of textual descriptions. The largest chest X-ray VQA dataset to date, CheXinstruct [1], contains 8.5M QA-pairs with images from multiple data sources. However, compared to our dataset, its questions and answers are less diverse, being purely template-based and derived from dataset annotations instead of being directly conditioned on the reports. Additionally, none of the described datasets provide the level of detail and annotation richness found in our dataset, which includes bounding boxes, tags, and more detailed, multi-part answers that mirror radiology report sentences.

**Grounded Report Generation** While localization is not yet common for medical VQA tasks, grounded report generation, i.e. predicting radiology reports with bounding boxes is gaining popularity. Notable examples include MAIRA-2 [18], trained on reports manually annotated with bounding boxes and MedTrinity-25M [7], a large-scale public dataset with automatically generated reports with bounding boxes. ChEX [2] is another model producing textual answers with bounding boxes. While being conditioned on textual prompts, ChEX does not support VQA tasks.

**Scene Graph Construction for Chest X-Rays** During our VQA dataset construction, we automatically derive scene graphs from radiology reports. A similar approach is employed by Chest ImaGenome [15], [17], [19], which uses rule-based information extraction, and RadGraph [20], which uses a relation extraction model. In contrast, our approach leverages LLM-based extraction with semantic entity mapping, enabling more comprehensive graph construction. Notably, our method defines a larger set of (localized) regions (257) and findings (221) compared to Chest ImaGenome (29 regions, 53 findings), making it a more robust foundation for VQA tasks.

## 3 The CXR-QBA Dataset

We present our dataset *CXR-QBA*, a large-scale *chest X-ray (CXR)* VQA dataset derived from *MIMIC-CXR* [14], [15], [21], consisting of more than 42M QA-pairs. As shown in Fig. 1, each QA sample (for a given chest X-ray) consists of a *question (Q)*, a *bounding box (B)* supplemented *answer (A)*, and additional tags (e.g. for regions, findings, certainties, and more).

To build our dataset, we propose an automatic pipeline highlighted in Fig. 2. More specifically, we first construct (visually grounded) scene graphs based on the MIMIC-CXR radiology reports using LLM-based information extraction, semantic concept mapping, and localization models (Sec. 3.1). These scene graphs provide a structured description of the study, including sentences (derived from the report) for individual observations. They serve as a data source for our question-answer generation, where we utilize both template-based answers and answers derived from the rewritten report sentences (Sec. 3.2). Finally, we automatically assess the quality of question-answer pairs using LLM-based evaluations (Sec. 3.3). Further details are provided in Appendices D and E.

## 3.1 Scene Graph Construction

Given a MIMIC-CXR study with a radiology report and accompanying CXRs, we construct a scene graph (Appendix D.1) consisting of sentence nodes, observation nodes, region nodes, an indication node, and edges between them. Sentence nodes are directly extracted from the reports, containing the raw sentences and their identified section names. Observation nodes represent individual aspects described in the report's `FINDINGS` or `IMPRESSION` section, containing (i) a textual description, (ii) bounding boxes for associated CXR images and (iii) additional tags, such as positivity, certainty, laterality, regions, and finding classes. Region nodes are created for mentioned anatomical structures and key regions. The indication node contains information from the `INDICATION` section, including a textual description and an individual observation node, derived from the `FINDINGS` and `IMPRESSION` sections, that can act as an answer to the indication. We construct these scene graphs in three steps: (a) region localization, (b) information extraction and (c) building the graphs using entity mapping. We refer to Appendix E.1 for details.

**Region Localization**  The bounding boxes in our scene graphs (and the derived QA-pairs) are based on fine-grained anatomical structures, allowing us to localize associated findings very precisely. We use the CXAS [22], [23] model to predict segmentation masks of 158 anatomical structures on the 377 110 CXRs from MIMIC-CXR-JPG [15], [24], [25]. Additionally, we use the bounding boxes provided by the Chest ImaGenome [15], [17], [19] dataset, which are provided for 29 anatomical structures in most frontal images of MIMIC-CXR. Next, we derive a total of 257 localized anatomical structures based on combinations (e.g. intersections, unions, super bounding boxes, etc.) of the available masks and bounding boxes. Finally, we discard any masks or boxes that are too small and derive bounding boxes from the segmentation masks. Note that we define 53 further regions/structures that are either non-localized (e.g. interstitial) or for which we do not have bounding boxes, leading to a total of 310 structures/regions.

**Information Extraction**  We use the 227 827 free-text radiology reports provided by MIMIC-CXR as the main source of information for our scene graphs. Using the Llama 3.1 70B [26] model with few-shot prompting, we extract the relevant information (tags and textual descriptions) in three steps. First, we extract individual sentences from the reports and detect their sections. Next, we extract information about the `INDICATION` section and detect which `FINDINGS` or `IMPRESSION` sentences may provide information related to the indication. Finally, we extract individual observations described in the `FINDING/IMPRESSION` sentences.

**Building Scene Graphs using Entity Mapping**  Given the extracted information from the reports and the computed bounding boxes, we now construct the final scene graph. Therefore, we first map extracted tags to pre-defined sets of values, our reference definitions. This assures high quality and consistency of the scene graphs and enables mapping of observations to the extracted bounding boxes. The reference definitions are based on tags used in other datasets (including PadChest [27] and Chest ImaGenome [19]) as well as SNOMED-CT [28]) and have been verified by clinical experts. They include synonym lists, hierarchies, and relationships. For more robust mapping, we utilize the BioLORD [29] model as a sentence transformer and identify the closest matching concept based on their semantic embeddings. Additionally, we try to fill in missing information where possible, such as inferring the region from an identified finding. Finally, we build a tree of region nodes (using the reference data) and attach the indication information extracted from the report.

## 3.2 Question-Answer Generation

We generate question-answer pairs (Appendix D.2) using a template-based approach based on the information available in the scene graphs, incorporating the textual descriptions from the observation nodes – which have been derived directly from the report – to provide diverse and fine-grained answers. Each answer may consist of multiple answer parts (as shown in Fig. 3 and Appendix A), each describing an individual aspect of the answer with its own sentence, bounding boxes, and tags. We categorize answer parts into three types: (i) main-answers, (ii) details, and (iii) related-information, allowing for controlled answer granularity. Answer parts are generated either from templates using scene graph information or directly from observation nodes (Appendix E.2). Answer parts may also be structured hierarchically, where we use parent-child edges from the scene graph.

To generate the question-answer pairs, we employ different strategies for the four types of questions (shown in Fig. 1):

1. **Indication**: We use the paraphrased indication as the question and create the answer based on the indication node in the scene graph, answering the indication based on information in the FINDINGS and IMPRESSION sections.

2. **Study abnormality**: We generate study-level questions using 13 different templates, with answer parts (Fig. 3) based on (filtered) observation nodes.

3. **Region abnormality**: We generate questions about individual regions using 6 different templates, considering any region mentioned and additionally randomly sampling non-mentioned regions for balancing.

4. **Finding**: We generate questions about individual findings using 7 different templates, considering any finding mentioned and additionally randomly sampling non-mentioned findings for balancing.

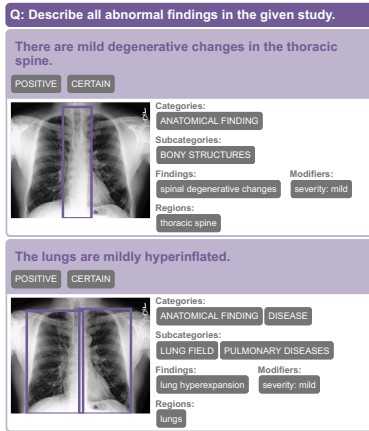

Figure 3: Answer with multiple parts for different aspects, each with a sentence, tags, and boxes.

### 3.3 Quality Assessment

The dataset construction procedure described so far allows us to automatically generate large amounts of QA-pairs. However, in each of the steps, errors may be introduced, affecting the overall quality of the datasets. For example, errors during information extraction could lead to incorrect tags, therefore leading to incorrectly filled answer templates or incorrectly selected observation nodes for answers.

In order to identify and filter such cases, we employ an automatic quality assessment strategy using an LLM as a judge. More specifically, we use Llama 3.1 8B [26] to rate question and answers by the following five criteria: *entailment* (does the answer factually align with the original report?), *relevance* (is the answer relevant to the question?), *completeness* (is the answer missing something?), as well as *question and answer clarity* (is the question/answer clear and grammatically correct?). Additionally, we assess the quality of the used scene graphs by identifying missing information (e.g. missing tags or localization) or issues during the construction process. Finally, we algorithmically combine these individual assessments to compute an overall quality rating as one of *A++, A+, A, B, C, D*, or *not rated* (see Appendix D.3). Based on these ratings, we propose two subsets, one for pre-training and one for fine-tuning. We exclude all non-frontal images from these datasets, as the localization quality on these images is comparatively low due to limitations in the localization models. All QA-pairs with a grade of A or better are labeled as *fine-tuning grade*, resulting in 7.5M pairs, while samples with grade B or better are considered *pre-training grade*, resulting in 31.2M pairs.

## 4 Evaluation and Analysis

### 4.1 Evaluation of the Scene Graphs

We evaluate our scene graphs by comparing their tags and bounding boxes to hand-labeled expert annotations on MIMIC-CXR, using the scene graphs from Chest ImaGenome [17] as a baseline. First, we evaluate the plausibility of finding tags by comparing study-level labels derived from our scene graphs to two reference annotation sets: the radiologist annotations in MIMIC-CXR-JPG v.2.1.0 [24] with 13 CheXpert [33] classes and the CXR-LT 2024 [15], [30], [34] gold-standard dataset (task 2 test set) with 12 additional rare (long-tail) classes. As shown in Tab. 2a, our approach (slightly) outperforms Chest ImaGenome, with strong improvements (20%) on long-tail classes, demonstrating the value of our fine-grained finding tags (221 classes) in capturing nuanced study details. To evaluate the accuracy of finding bounding boxes, we compare them with annotations from MS-CXR [15], [31], [35] (6 classes) and REFLACX [15], [32], [36] (18 classes). We compute study-level pixel masks for each finding as the union of all bounding boxes from positive observation nodes that contain the specific finding tag. We calculate pixel-level Intersection-over-Union (IoU), Intersection-over-Prediction (IoP), and Intersection-over-Target (IoT) for each finding class, considering only image

Table 2: Evaluation of our scene graphs, comparing finding tags (a) and associated bounding boxes (b) to expert annotations on MIMIC-CXR subsets, with 95% confidence intervals (bootstrapping, $n = 1000$). Compared to Chest ImaGenome's [17] scene graphs, we achieve competitive or superior performance, showing that our construction process yields plausible scene graphs.

(a) Evaluation of finding tags against 13 CheXpert (CXP) classes from the MIMIC-CXR-JPG test set and 25 classes, 13 CXP and 12 long-tail (LT) classes, from the CXR-LT 2024 gold standard dataset (Appendix C.2). We report the Matthews Correlation Coefficient (MCC) macro-averaged over different finding subsets (CXP-5, CXP-7, CXP-13, LT) and micro-averaged. Compared to Chest ImaGenome, we produce slightly more accurate tags, performing especially well on long-tail classes, highlighting the importance of our fine-grained tags.

| Classes | MIMIC-CXR-JPG [24] Test [MCC] | | | | CXR-LT 2024 [30] Gold [MCC] | | | | |
| | CXP-5 | CXP-7 | CXP-13 | Micro | CXP-7 | CXP-13 | LT-only | CXR-LT | Micro |
| --- | --- | --- | --- | --- | --- | --- | --- | --- | --- |
| Ours (scene graphs) | **0.8** [0.77, 0.82] | **0.81** [0.79, 0.84] | **0.69** [0.67, 0.71] | **0.71** [0.69, 0.73] | **0.65** [0.61, 0.69] | **0.57** [0.54, 0.6] | **0.71** [0.67, 0.74] | **0.64** [0.61, 0.66] | **0.67** [0.65, 0.69] |
| Chest ImaGenome | 0.78 [0.75, 0.81] | 0.8 [0.78, 0.83] | 0.66 [0.64, 0.69] | 0.67 [0.65, 0.68] | **0.65** [0.61, 0.68] | 0.56 [0.54, 0.59] | 0.59 [0.55, 0.63] | 0.58 [0.55, 0.6] | 0.64 [0.62, 0.66] |

(b) Evaluation of finding bounding boxes against 6 finding classes from MS-CXR and 18 classes from REFLACX (Appendix C.3). We report the pixel-level Intersection-over-Union (IoU), Intersection-over-Prediction (IoP), and Intersection-over-Target (IoT), each thresholded at 30%, and micro-averaged. Compared to Chest ImaGenome, our bounding boxes are better matching the hand-labeled boxes, especially leading to smaller and more precise boxes (larger IoP), which we assume is due to our more fine-grained region annotations.

| | MS-CXR [31] | | | REFLACX [32] | | |
| | [IoU@30] | [IoP@30] | [IoT@30] | [IoU@30] | [IoP@30] | [IoT@30] |
| --- | --- | --- | --- | --- | --- | --- |
| Ours (scene graphs) | **0.51** [0.47, 0.54] | **0.56** [0.52, 0.6] | 0.94 [0.92, 0.96] | **0.45** [0.44, 0.47] | **0.54** [0.53, 0.56] | 0.87 [0.86, 0.88] |
| Chest ImaGenome | 0.45 [0.42, 0.49] | 0.48 [0.45, 0.52] | **0.98** [0.97, 0.99] | 0.42 [0.4, 0.43] | 0.46 [0.44, 0.47] | **0.95** [0.94, 0.96] |

pairs with positive predictions and targets. Thresholding at 30% IoU/IoP/IoT, we micro-average the results, reporting the percentage of accurately localized finding-boxes in Tab. 2b. On the IoU metric, our scene graphs perform slightly better than the ones from Chest ImaGenome. The low IoP values indicate that bounding boxes are often too large, but high IoT values suggest that they generally cover the finding boxes well. This discrepancy arises because bounding boxes are derived from anatomical regions mentioned in reports, whereas hand-labeled annotations are more precise. Notably, our approach produces more precise boxes (higher IoP) than Chest ImaGenome, likely due to our large number of fine-grained region annotations (257 region classes).

Our analysis confirms that our scene graphs contain plausible finding tags and bounding boxes, with competitive or better quality than Chest ImaGenome. The bounding box quality, in turn, validates the plausibility of our region tags. Overall, our construction process yields high-quality scene graphs, making them a reliable foundation for generating QA samples.

## 4.2 Quality of the QA-Samples

We assessed the quality of our 42.2M QA-pairs using an LLM-as-a-judge approach (Sec. 3.3). Results are shown in Fig. 4a. We found that 18.6% were fine-tuning grade, 58.8% were pre-training grade, and 22.6% were marked for exclusion. Notably, 85% of individual main answers were rated A or higher. We also analyzed the main causes of ratings (Fig. 4b) and found that A+ samples were limited by minor incompleteness (minor details missing), A samples by minor entailment aspects (facts not explicitly mentioned in the report), while B samples were restricted by issues with region/finding/localization extraction, completeness, and text clarity. Ratings C were caused by major incompleteness or extraction issues, ratings D by contradicting entailments, while non-rated samples where due to the LLM-judge not producing parsable outputs. Using a larger LLM judge (Llama 3.1 70B), tested on a subset, reduced exclusions by 20%, but we opted for the smaller model (Llama 3.1 8B) to reduce computational requirements (we refer to Appendix C.1 for further details). Our analysis shows that even pre-training grade samples provide factually accurate answers with minor flaws, making them suitable for pre-training purposes.

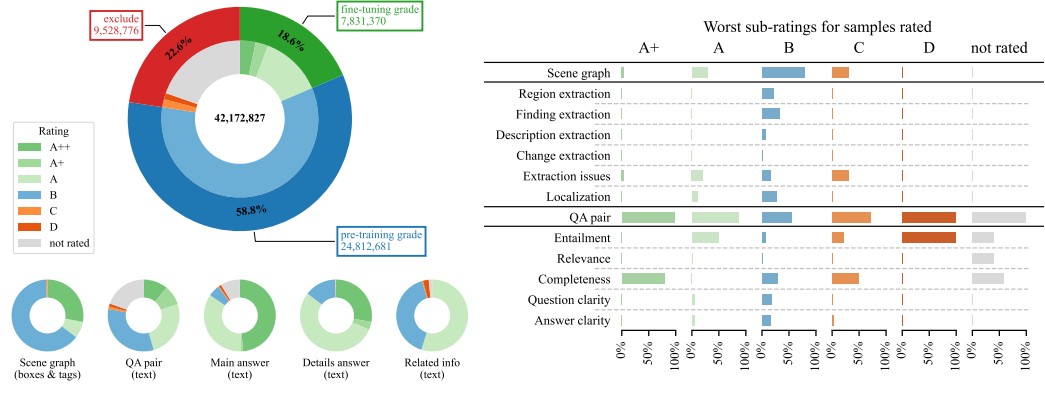

(a) Overall rating (top) and sub-ratings (bottom).

(b) Reasons for ratings.

Figure 4: Results of quality assessment (Sec. 3.3). We identified a significant amount of fine-tuning grade samples, while even pre-training grade samples provide factually accurate answers, especially having high quality main answers.

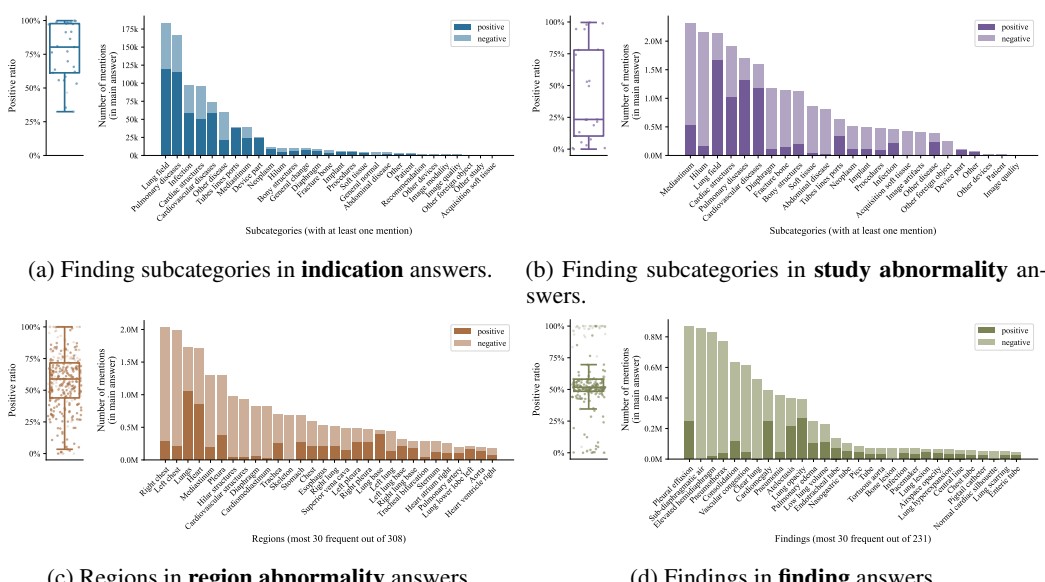

(a) Finding subcategories in **indication** answers.

(b) Finding subcategories in **study abnormality** answers.

(c) Regions in **region abnormality** answers.

(d) Findings in **finding** answers.

Figure 5: Distribution of tags (finding subcategories, regions, findings) mentioned in answers of different question types (indication, study abnormality, region abnormality, finding). We show their positive ratios, i.e. how often they are mentioned in positive versus in negative answers (**left**) and plot the number of positive and negative mentions of the most frequent tags (**right**). These fine-grained tags enable filtering and balancing the dataset or can be used as additional supervision.

### 4.3 Finding- and Region-Distribution in QA-Samples

Our answers include additional tags for findings (and their categories), regions, and answer positivity (positive or negative finding), enabling filtering and balancing for specific applications. For instance, undersampling negative answers can help mitigate model biases towards negative predictions. In Fig. 5 we analyze the distribution of these tags. We observe that indication questions tend to have more positive mentions (Fig. 5a) – as there is a specific indication to check for – while study abnormality questions have more negative ones (Fig. 5b) – as many samples are negative overall. In region abnormality questions, most regions are mentioned slightly more often with positive than with negative findings (Fig. 5c), while for finding questions mentions are mostly balanced (Fig. 5d). This shows the success of our balanced region/finding sampling used for these two question types.

## 4.4 Answer Characteristics

Our QA-samples provide detailed free-text answers consisting of one or even multiple sentences (i.e. answer parts). In Fig. 6, we analyze the distribution of lengths of these answers and study differences between types of answers or questions. The median answer length is 14 words, with similar lengths for most question types except for indication questions, where answers are much longer (46 words). We also observe that related information answers are much longer (22 words) than main answers (9 words) or details answers (7 words), which is expected as they can provide a lot additional context to the answers. Answers describing positive findings are typically very long (18 words), considerably longer than negative finding answers (10 words). This highlights that our dataset provides nuanced finding description in their answers, following the level of detail typically present in radiology reports.

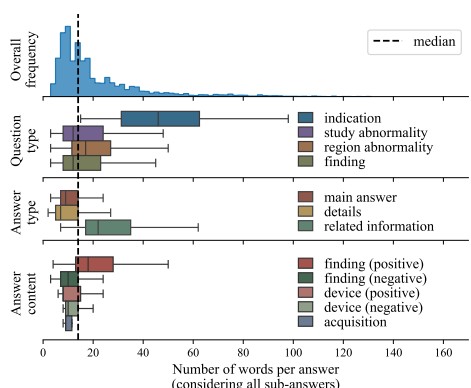

Figure 6: Distribution of answer lengths. We provide nuanced answers with detailed free-text finding descriptions.

## 5 Structured VQA Task

To demonstrate the utility of our dataset, we introduce a structured Visual Question Answering (VQA) task. This task requires models to generate free-text answers accompanied by bounding boxes and tags (e.g., findings, regions). Given a chest X-ray and a free-text question, the model must output such structured answers to respond to the query. We refer to Appendix F for further details.

**Sequence Formatting for Structured VQA** We implement a proof-of-concept model based on the Llava architecture [37], using Rad-DINO [38] for image encoding and the Llama 3.2 3B [26] language model connected via an MLP projection layer. Our CXR-QBA dataset provides the necessary targets, which we format into sequences using XML-style structures and special tokens to represent tags and bounding boxes (converted to relative coordinates following [18]). We then fine-tune the model for one epoch on 1M QA-pairs (MIMIC-CXR train split).

**RadStrucVQA Metric** For evaluation, we introduce the *RadStrucVQA* metric, which closely follows the RadFact [18] metric introduced for radiology report generation but is generalized to structured VQA. Like RadFact, we identify whether individual predicted answer parts are entailed with target answer parts and vice-versa, in our casing using Llama 3.1 8B. For entailed pairs, we compute whether they are visually grounded, i.e. whether their bounding boxes are precise enough considering their references, and whether finding and region tags are correctly reported. This is conducted bi-directionally, using either the targets as references for the predictions or vice-versa, resulting in precision or recall scores, respectively. More details can be found in Appendix F.2.

Table 3: Results on the structured VQA task, with 95% confidence intervals (bootstrapping, $n = 1000$). **Left**: Our model trained on this task. **Right**: MAIRA-2 with adapted prompt. Our dataset enables training vision-language models to predict logically correct, visually grounded answers, supplemented by tags that facilitate thorough analysis of the model's predictions.

|  |  | Ours | MAIRA-2 |
|---|---|---|---|
| RadFact* | Logical Prec. | **0.76** [0.75, 0.76] | 0.25 [0.25, 0.26] |
|  | Logical Rec. | **0.75** [0.74, 0.76] | 0.64 [0.63, 0.65] |
|  | Grounding Prec. | **0.87** [0.87, 0.88] | 0.69 [0.67, 0.71] |
|  | Grounding Rec. | **0.89** [0.88, 0.89] | 0.12 [0.11, 0.12] |
| RadStrucVQA (Tags) | Finding Prec. | **0.68** [0.67, 0.69] | – |
|  | Finding Rec. | **0.66** [0.66, 0.67] | – |
|  | Finding-pos Prec. | **0.41** [0.40, 0.43] | – |
|  | Finding-pos Rec. | **0.26** [0.25, 0.27] | – |
|  | Region Prec. | **0.67** [0.66, 0.68] | – |
|  | Region Rec. | **0.66** [0.65, 0.67] | – |

*Our RadStrucVQA implementation.

**Results** We evaluate our model on our fine-tuning grade dataset (MIMIC-CXR test split) and compare it to MAIRA-2 [18], a model for grounded report generation trained partially on MIMIC-

CXR, i.e. on the same images as our model. We use the frozen MAIRA-2 model, but adapt its prompt to answer specific questions instead of generating full reports. Results are shown in Tab. 3. Our model achieves high scores in both textual content (logical) and grounding metrics, demonstrating effective training on our dataset. As expected, MAIRA-2 performed lower on all metrics, but achieved $85\%$ of our models logical recall, suggesting it captured most relevant information while also including extraneous details (lower precision). This aligns with its training objective of comprehensive reporting but also indicates that our dataset's answers do not contradict with MAIRA-2's predictions, further confirming the quality of our dataset. MAIRA-2's grounding precision significantly exceeded its recall, because it was trained to predict bounding boxes only for positive findings. Our model successfully predicts finding and region tags in most cases. However, performance drops when focusing solely on positive findings (`finding-pos`), indicating potential underprediction possibly due to our training procedure or limitations in the pre-trained components. While further analysis would be required, this may indicate problems that could also lead to flaws in textual answers. Importantly, our datasets detailed tags enable fine-grained analysis of such issues while also enabling potential solutions like data filtering or balancing, making it well-suited for complex training scenarios.

## 6  Discussion and Conclusion

### 6.1  Use Cases and Impact

Our dataset is particularly well-suited for structured VQA on CXRs (Sec. 5). Additionally, its versatility also supports classical VQA tasks or grounded VQA without structure, while its large size and detailed answers make it a valuable resource for pre-training vision-language models. The accompanying tags further enable filtering and balancing of the dataset to suite specific needs.

Use cases are, however, not limited to VQA tasks. Our fine-grained scene graphs with bounding boxes, textual descriptions, and tags can serve as a versatile data source for various purposes. For instance, they can be leveraged to create customized datasets for grounded report generation or VQA, or even as a direct training source for graph generation models to predict scene graphs on unseen chest X-rays, enabling the creation of even larger datasets. Furthermore, the bounding boxes and tags provided with the scene graphs can be used for longitudinal analysis, including region-level examination. Finally, they can be used to train models for pathology localization or classification, providing fine-grained and long-tail diagnosis targets that are often lacking in existing datasets.

### 6.2  Limitations

Our dataset was automatically constructed, relying on models and templates instead of human annotations. While this enables the generation of a large number of QA-pairs, it may also introduce potential errors and biases. We apply (automatic) quality assessments to mitigate these risks, but users should still be aware that the dataset may contain inaccuracies and should exercise caution, especially when using it for critical applications. Most importantly, we strictly advise against using this dataset as the sole source for fine-tuning or evaluating models used in clinical practice. Furthermore, our template-based approach may limit the diversity of the dataset and may potentially introduce grammatical errors. However, we partially mitigate these issues by incorporating answers derived from actual report sentences and through our quality assessment measures. Additionally, our approach focuses on individual chest X-ray studies, excluding longitudinal, differential questions, and other (imaging) modalities. Future extensions could build upon this work, generalizing our approach to broader question types and modalities. Finally, our work relies on LLMs for information extraction and quality assessment. While we only use medium to small models, these still require substantial computational resources for dataset creation, particularly compared to template-based methods.

### 6.3  Conclusion

We proposed a novel approach to constructing a large-scale CXR VQA dataset using automatic scene graph construction and question-answer generation, resulting in CXR-QBA, a dataset of 42 M QA-pairs. We hope that our dataset will serve as a valuable resource for researchers and practitioners, driving advancements in medical imaging and vision-language understanding.

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

# A   Example QA-Pairs from our Dataset

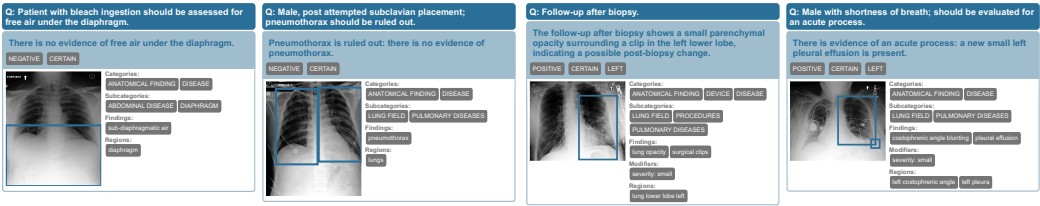

(a) Negative.                    (b) Positive.

Figure 7: Examples of **indication** questions. Questions are based on the paraphrased `INDICATION` section while each main answer is generated based on the indication node from the scene graph (using information from the `FINDINGS` and `IMPRESION` sections).

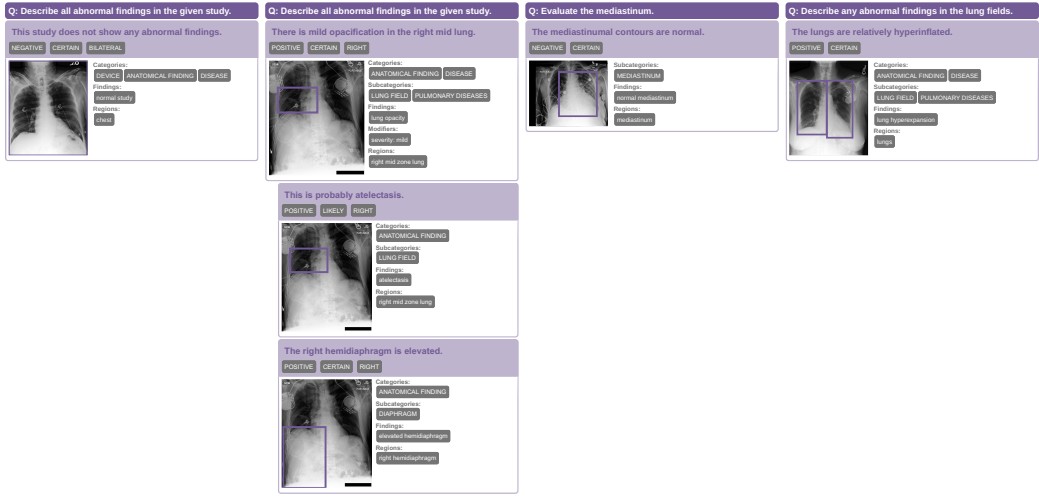

(a) Abnormality descriptions (study-level or category-level).

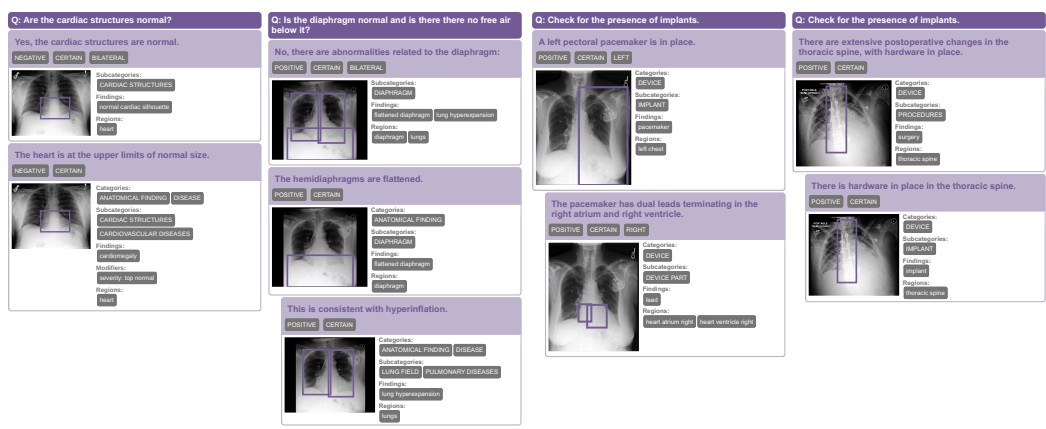

(b) Abnormality assessment (category-level).           (c) Device descriptions.

Figure 8: Examples of **study abnormality** questions. Questions are based on one of 13 templates. Answers may consist of several answer parts, where each describes an individual aspect (about the overall study or a finding category). Individual answer parts are constructed based on observation nodes, filtered based on finding categories relevant to the question, where individual answer parts may be organized hierarchically (indicated by indentations) based on parent-child edges in the scene graph. Additionally assessment answers (b) start with a template-based yes/no answer.

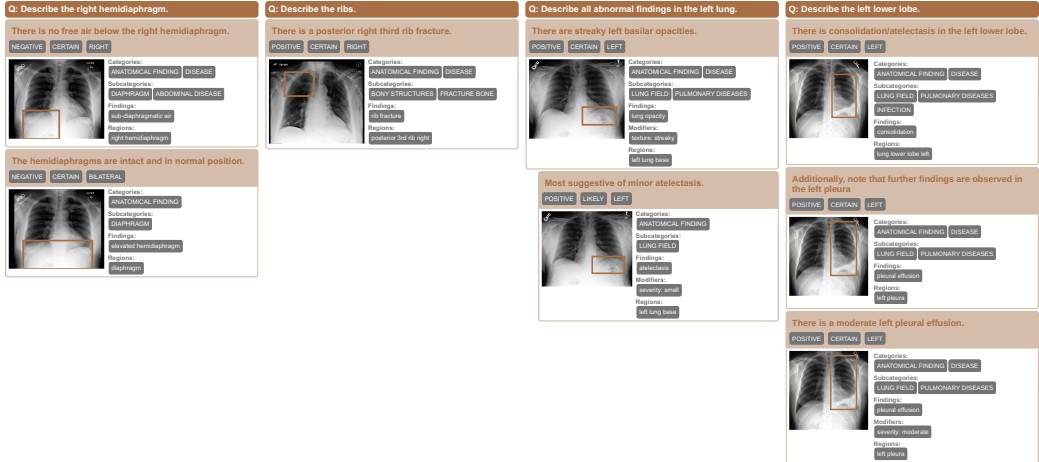

(a) Region description.

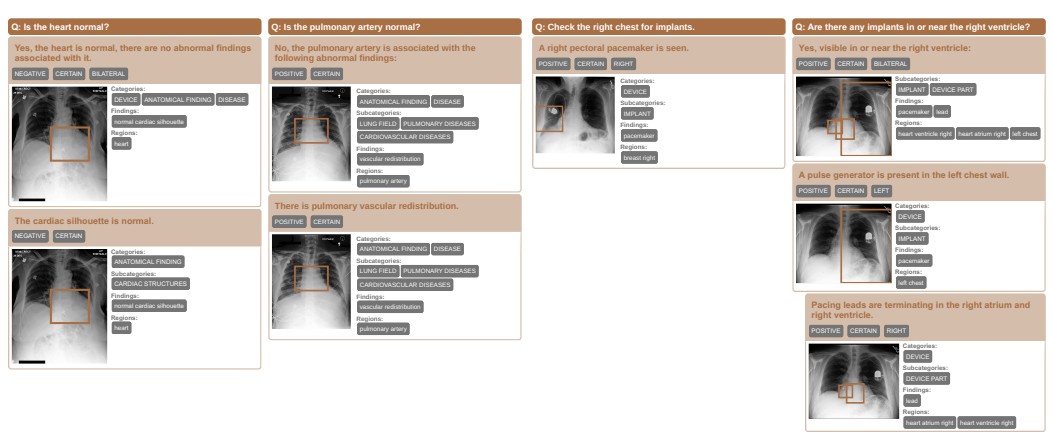

(b) Region assessment.

(c) Region devices.

Figure 9: Examples of **region abnormality** questions. Questions are based on one of 6 templates. Answers may consist of several answer parts, where each describes an individual aspect (about the region). Individual answer parts are constructed based on observation nodes relevant to the region, where individual answer parts may be organized hierarchically (indicated by indentations) based on parent-child edges in the scene graph. Additionally assessment answers (b) start with a template-based yes/no answer. Some templates also ask specifically about devices in the region (c).

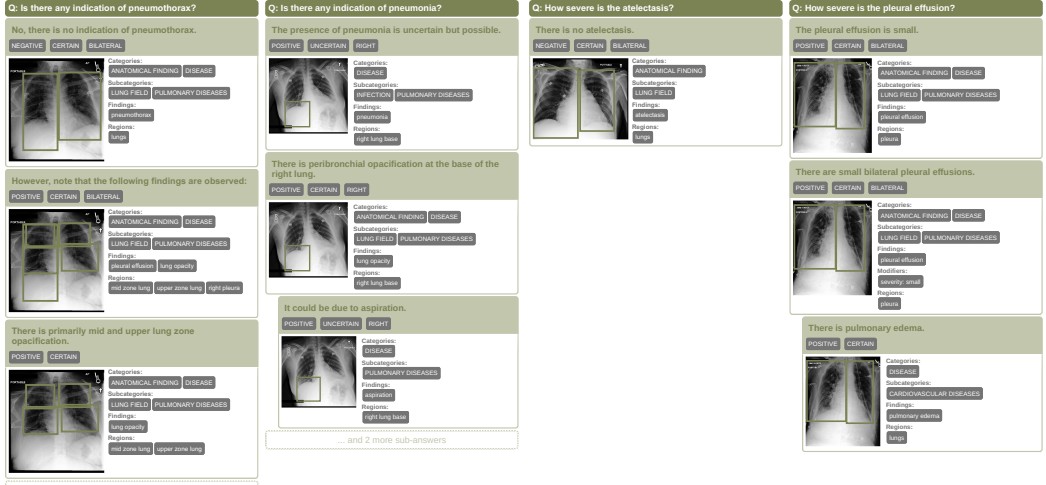

(a) Finding assessment.            (b) Finding description.

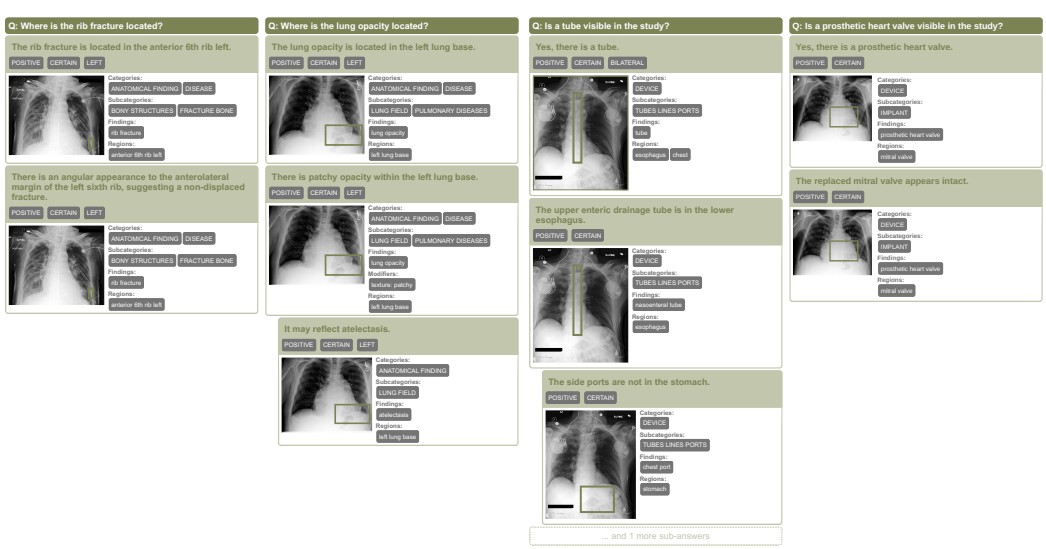

(c) Finding location.            (d) Device.

Figure 10: Examples of **finding** questions. Questions are based on one of 7 templates. Answers start with a template-based answer part to identify the finding presence (a), provide a severity summary (b), describe the location (c), or presence of a device (d). Additional details may be provided in answer parts based on observation nodes relevant to the finding, where individual answer parts may be organized hierarchically (indicated by indentations) based on parent-child edges in the scene graph.

 # B  Finding- and Region-Distribution in QA-Samples

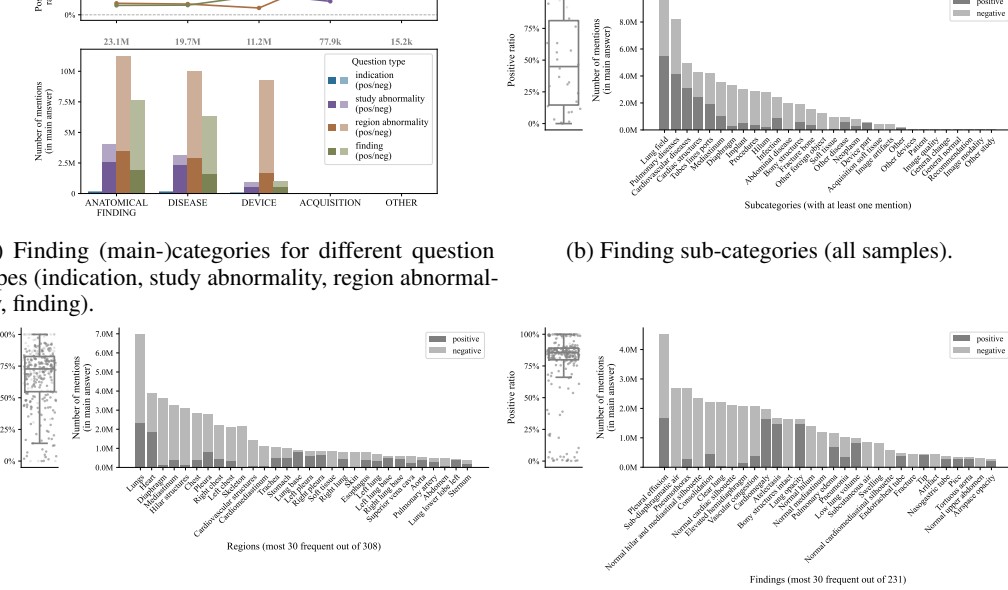

(a) Finding (main-)categories for different question types (indication, study abnormality, region abnormality, finding).

(b) Finding sub-categories (all samples).

(c) Regions (all samples).

(d) Findings (all samples).

Figure 11: Tags (finding main- and sub-categories, regions, findings) mentioned in answers. We show their positive ratios (**top/left**), i.e. how often they are mentioned in positive versus in negative answer parts and plot the number of positive and negative mentions of the most frequent tags (**bottom/right**).

# C Evaluation Details

## C.1 QA Evaluation: Comparison of LLM-Raters

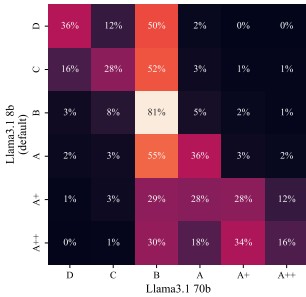

Figure 12: Confusion matrix comparing the assigned quality ratings between using Llama3.1 8b (default) and Llama3.1 70b as an LLM-judge (see Secs. 3.3 and 4.2). In most cases, ratings differ only slightly. Most importantly, low-quality samples (as rated by Llama3.1 70b) are almost never assigned to fine-tuning grades (A or higher) by Llama3.1 8b. We thus decided to use Llama3.1 8b as our default rater, as it is much more computationally efficient.

## C.2 Scene Graph Evaluation: Finding Tags

Table 4: Evaluation of finding tags against the 13 CheXpert (CXP) classes from the MIMIC-CXR-JPG test set (Sec. 4.1). We show finding-level scores, macro-averages over subsets and the micro-average, with 95% confidence intervals (bootstrapping, $n = 1000$).

| | MIMIC-CXR-JPG [24] Test | | | | | | | |
|---|---|---|---|---|---|---|---|---|
| | [Precision] | | [Recall] | | [F1] | | [MCC] | |
| | Ours | Chest ImaG. | Ours | Chest ImaG. | Ours | Chest ImaG. | Ours | Chest ImaG. |
| **Findings in CXP-5, CXP-7, and CXP-13** | | | | | | | | |
| Atelectasis | **0.82** [0.78, 0.87] | 0.78 [0.73, 0.83] | **0.99** [0.97, 1.0] | **0.99** [0.98, 1.0] | **0.9** [0.87, 0.93] | 0.88 [0.84, 0.9] | **0.84** [0.8, 0.88] | 0.81 [0.77, 0.85] |
| Cardiomegaly | 0.64 [0.58, 0.7] | **0.67** [0.61, 0.74] | **0.85** [0.78, 0.9] | 0.82 [0.75, 0.87] | 0.73 [0.67, 0.78] | **0.74** [0.68, 0.79] | 0.61 [0.54, 0.68] | **0.63** [0.56, 0.69] |
| Consolidation | **0.83** [0.73, 0.91] | 0.77 [0.68, 0.86] | 0.87 [0.79, 0.94] | **0.93** [0.86, 0.99] | **0.85** [0.78, 0.91] | 0.84 [0.78, 0.9] | **0.83** [0.75, 0.9] | **0.83** [0.76, 0.89] |
| Edema | **0.94** [0.9, 0.98] | 0.9 [0.85, 0.95] | **0.8** [0.73, 0.86] | **0.8** [0.74, 0.86] | **0.86** [0.82, 0.9] | 0.85 [0.8, 0.89] | **0.83** [0.77, 0.87] | 0.8 [0.74, 0.85] |
| Pleural Effusion | **0.9** [0.86, 0.93] | 0.86 [0.82, 0.9] | **0.98** [0.96, 1.0] | 0.97 [0.94, 0.99] | **0.94** [0.92, 0.96] | 0.91 [0.89, 0.94] | **0.89** [0.85, 0.92] | 0.85 [0.8, 0.89] |
| **Findings in CXP-7 and CXP-13** | | | | | | | | |
| Pneumonia | **0.92** [0.87, 0.96] | 0.89 [0.84, 0.94] | 0.94 [0.89, 0.97] | **0.95** [0.91, 0.98] | **0.93** [0.89, 0.96] | 0.92 [0.89, 0.95] | **0.91** [0.87, 0.95] | 0.9 [0.86, 0.94] |
| Pneumothorax | 0.78 [0.64, 0.89] | **0.79** [0.66, 0.91] | 0.84 [0.71, 0.95] | **0.89** [0.78, 0.98] | 0.8 [0.69, 0.89] | **0.84** [0.74, 0.92] | 0.79 [0.68, 0.88] | **0.83** [0.72, 0.91] |
| **Findings in CXP-13** | | | | | | | | |
| Enlarged Cardiom. | 0.51 [0.37, 0.65] | **0.61** [0.41, 0.8] | **0.39** [0.29, 0.51] | 0.23 [0.13, 0.33] | **0.44** [0.34, 0.55] | 0.33 [0.2, 0.45] | **0.39** [0.28, 0.51] | 0.33 [0.19, 0.45] |
| Lung Lesion | 0.17 [0.12, 0.22] | **0.68** [0.56, 0.79] | 0.81 [0.69, 0.91] | **0.87** [0.76, 0.95] | 0.28 [0.21, 0.35] | **0.76** [0.66, 0.84] | 0.25 [0.17, 0.32] | **0.74** [0.64, 0.83] |
| Lung Opacity | **0.62** [0.56, 0.69] | 0.28 [0.24, 0.31] | 0.83 [0.77, 0.89] | **1.0** [1.0, 1.0] | **0.71** [0.65, 0.76] | 0.43 [0.39, 0.48] | **0.61** [0.54, 0.68] | 0.2 [0.17, 0.23] |
| Pleural Other | **0.54** [0.36, 0.71] | 0.3 [0.19, 0.42] | 0.87 [0.7, 1.0] | **0.92** [0.76, 1.0] | **0.67** [0.49, 0.8] | 0.45 [0.31, 0.58] | **0.67** [0.51, 0.79] | 0.5 [0.37, 0.6] |
| Fracture | **0.67** [0.54, 0.79] | 0.6 [0.47, 0.73] | **0.92** [0.82, 1.0] | 0.82 [0.68, 0.93] | **0.77** [0.67, 0.86] | 0.69 [0.58, 0.79] | **0.77** [0.67, 0.85] | 0.68 [0.56, 0.78] |
| Support Devices | 0.61 [0.56, 0.66] | **0.62** [0.56, 0.67] | **0.98** [0.96, 1.0] | 0.83 [0.77, 0.88] | **0.75** [0.71, 0.79] | 0.71 [0.66, 0.75] | **0.63** [0.59, 0.68] | 0.54 [0.48, 0.61] |
| **Macro-averages** | | | | | | | | |
| CheXpert-5 (CXP-5) | **0.83** [0.79, 0.85] | 0.8 [0.77, 0.83] | **0.9** [0.87, 0.92] | **0.9** [0.88, 0.92] | **0.85** [0.83, 0.87] | 0.84 [0.82, 0.86] | **0.8** [0.77, 0.82] | 0.78 [0.75, 0.81] |
| CheXpert-7 (CXP-7) | **0.83** [0.8, 0.86] | 0.81 [0.78, 0.84] | 0.89 [0.87, 0.92] | **0.91** [0.88, 0.93] | **0.86** [0.83, 0.88] | 0.85 [0.83, 0.87] | **0.81** [0.79, 0.84] | 0.8 [0.78, 0.83] |
| CheXpert-13 (CXP-13) | **0.69** [0.66, 0.71] | 0.67 [0.65, 0.7] | **0.85** [0.83, 0.87] | **0.85** [0.82, 0.87] | **0.74** [0.72, 0.76] | 0.72 [0.7, 0.74] | **0.69** [0.67, 0.71] | 0.66 [0.64, 0.69] |
| Micro | **0.68** [0.66, 0.7] | 0.63 [0.61, 0.65] | **0.89** [0.87, 0.9] | 0.88 [0.86, 0.89] | **0.77** [0.75, 0.79] | 0.73 [0.72, 0.75] | **0.71** [0.69, 0.73] | 0.67 [0.65, 0.68] |

Table 5: Evaluation of finding tags against the 13 CXP and 12 long-tail (LT) classes from the CXR-LT 2024 gold standard dataset (Sec. 4.1). We show finding-level scores, macro-averages over different subsets and the micro-average, with 95% confidence intervals (bootstrapping, $n = 1000$).

| | CXR-LT 2024 [30] Gold | | | | | | | |
| --- | --- | --- | --- | --- | --- | --- | --- | --- |
| | [Precision] | | [Recall] | | [F1] | | [MCC] | |
| | Ours | Chest ImaG. | Ours | Chest ImaG. | Ours | Chest ImaG. | Ours | Chest ImaG. |
| **Findings in CXP-5, CXP-7, CXP-13, and CXR-LT** | | | | | | | | |
| Atelectasis | 0.55 [0.47, 0.62] | **0.56** [0.49, 0.61] | 0.82 [0.75, 0.88] | **0.99** [0.97, 1.0] | 0.66 [0.59, 0.71] | **0.71** [0.65, 0.76] | 0.48 [0.4, 0.57] | **0.59** [0.54, 0.65] |
| Cardiomegaly | 0.82 [0.76, 0.88] | **0.85** [0.79, 0.91] | **0.85** [0.79, 0.91] | 0.8 [0.73, 0.86] | **0.84** [0.79, 0.88] | 0.82 [0.77, 0.87] | **0.73** [0.66, 0.79] | 0.72 [0.65, 0.79] |
| Consolidation | **0.82** [0.73, 0.91] | 0.74 [0.63, 0.83] | 0.86 [0.76, 0.93] | **0.89** [0.8, 0.95] | **0.84** [0.77, 0.9] | 0.8 [0.73, 0.87] | **0.8** [0.72, 0.87] | 0.76 [0.67, 0.83] |
| Edema | **0.73** [0.63, 0.83] | 0.69 [0.6, 0.79] | 0.64 [0.55, 0.74] | **0.71** [0.61, 0.8] | 0.68 [0.6, 0.76] | **0.7** [0.62, 0.77] | 0.59 [0.5, 0.69] | **0.6** [0.5, 0.69] |
| Pleural Effusion | **0.82** [0.76, 0.87] | 0.78 [0.73, 0.84] | 0.93 [0.9, 0.97] | **0.97** [0.94, 0.99] | **0.87** [0.83, 0.91] | 0.87 [0.83, 0.9] | **0.77** [0.7, 0.82] | 0.76 [0.7, 0.82] |
| **Findings in CXP-7, CXP-13, and CXR-LT** | | | | | | | | |
| Pneumonia | **0.38** [0.19, 0.58] | 0.13 [0.07, 0.21] | 0.45 [0.25, 0.67] | **0.76** [0.55, 0.94] | **0.41** [0.22, 0.58] | 0.23 [0.13, 0.33] | **0.38** [0.19, 0.55] | 0.24 [0.13, 0.34] |
| Pneumothorax | **0.85** [0.73, 0.95] | 0.8 [0.7, 0.9] | 0.85 [0.72, 0.94] | **0.96** [0.89, 1.0] | 0.85 [0.75, 0.92] | **0.87** [0.8, 0.94] | 0.83 [0.72, 0.91] | **0.86** [0.78, 0.93] |
| **Findings in CXP-13 and CXR-LT** | | | | | | | | |
| Enlarged Cardiom. | 0.78 [0.57, 0.95] | **1.0** [1.0, 1.0] | **0.13** [0.07, 0.2] | 0.1 [0.05, 0.17] | **0.22** [0.13, 0.32] | 0.18 [0.1, 0.29] | 0.24 [0.13, 0.35] | **0.27** [0.19, 0.36] |
| Lung Lesion | 0.01 [0.0, 0.03] | **0.05** [0.0, 0.12] | 0.5 [0.0, 1.0] | **0.75** [0.0, 1.0] | 0.02 [0.0, 0.06] | **0.1** [0.0, 0.22] | 0.0 [-0.09, 0.1] | **0.18** [-0.02, 0.3] |
| Lung Opacity | **0.92** [0.87, 0.96] | 0.54 [0.49, 0.59] | 0.77 [0.72, 0.84] | **1.0** [1.0, 1.0] | **0.84** [0.8, 0.88] | 0.7 [0.66, 0.74] | **0.73** [0.66, 0.79] | 0.35 [0.29, 0.4] |
| Pleural Other | 0.11 [0.0, 0.24] | **0.23** [0.13, 0.34] | 0.19 [0.0, 0.43] | **1.0** [1.0, 1.0] | 0.14 [0.0, 0.29] | **0.38** [0.23, 0.51] | 0.1 [-0.05, 0.26] | **0.45** [0.34, 0.55] |
| Fracture | **0.89** [0.78, 0.98] | 0.84 [0.72, 0.94] | **0.91** [0.81, 0.98] | 0.82 [0.69, 0.93] | **0.9** [0.82, 0.95] | 0.83 [0.73, 0.9] | **0.89** [0.8, 0.95] | 0.81 [0.69, 0.89] |
| Support Devices | 0.92 [0.88, 0.96] | **0.93** [0.9, 0.97] | **0.93** [0.89, 0.96] | 0.83 [0.77, 0.88] | **0.92** [0.9, 0.95] | 0.88 [0.84, 0.91] | **0.83** [0.77, 0.88] | 0.75 [0.68, 0.82] |
| **Findings in LT-only, and CXR-LT** | | | | | | | | |
| Calcification of the Aorta | **0.95** [0.83, 1.0] | **0.95** [0.88, 1.0] | 0.43 [0.28, 0.58] | **0.93** [0.85, 1.0] | 0.6 [0.43, 0.73] | **0.94** [0.89, 0.99] | 0.61 [0.48, 0.73] | **0.94** [0.87, 0.99] |
| Emphysema | **0.58** [0.41, 0.74] | 0.54 [0.38, 0.69] | **0.81** [0.63, 0.95] | 0.81 [0.63, 0.95] | **0.68** [0.52, 0.8] | 0.65 [0.49, 0.77] | **0.66** [0.5, 0.79] | 0.63 [0.47, 0.75] |
| Fibrosis | **0.27** [0.15, 0.43] | 0.0 [0.0, 0.0] | **0.52** [0.31, 0.74] | 0.0 [0.0, 0.0] | **0.36** [0.21, 0.52] | 0.0 [0.0, 0.0] | **0.33** [0.17, 0.5] | 0.0 [0.0, 0.0] |
| Hernia | **1.0** [1.0, 1.0] | 0.86 [0.68, 1.0] | **0.9** [0.73, 1.0] | **0.9** [0.73, 1.0] | **0.95** [0.85, 1.0] | 0.88 [0.74, 0.97] | **0.95** [0.85, 1.0] | 0.87 [0.73, 0.97] |
| Infiltration | 0.15 [0.03, 0.3] | **0.38** [0.16, 0.67] | 0.33 [0.08, 0.7] | **0.55** [0.21, 0.88] | 0.21 [0.05, 0.39] | **0.44** [0.18, 0.69] | 0.19 [0.01, 0.39] | **0.44** [0.17, 0.69] |
| Mass | **0.54** [0.34, 0.74] | 0.28 [0.17, 0.4] | 0.78 [0.56, 0.94] | **0.89** [0.71, 1.0] | **0.64** [0.44, 0.78] | 0.42 [0.28, 0.56] | **0.63** [0.45, 0.77] | 0.46 [0.32, 0.58] |
| Nodule | **0.92** [0.79, 1.0] | 0.5 [0.36, 0.63] | 0.74 [0.57, 0.88] | **0.91** [0.78, 1.0] | **0.82** [0.68, 0.91] | 0.64 [0.51, 0.75] | **0.81** [0.68, 0.91] | 0.64 [0.51, 0.74] |
| Pleural Thickening | **0.78** [0.62, 0.92] | 0.33 [0.22, 0.45] | **1.0** [1.0, 1.0] | **1.0** [1.0, 1.0] | **0.88** [0.77, 0.96] | 0.5 [0.36, 0.62] | **0.88** [0.78, 0.96] | 0.54 [0.43, 0.64] |
| Pneumomediastinum | **0.94** [0.83, 1.0] | 0.88 [0.73, 0.97] | **0.84** [0.7, 0.96] | 0.84 [0.7, 0.96] | **0.89** [0.78, 0.96] | 0.86 [0.73, 0.94] | **0.88** [0.77, 0.95] | 0.85 [0.72, 0.93] |
| Pneumoperitoneum | **0.88** [0.72, 1.0] | 0.82 [0.65, 0.96] | 0.96 [0.85, 1.0] | **1.0** [1.0, 1.0] | **0.92** [0.81, 1.0] | 0.9 [0.79, 0.98] | **0.91** [0.8, 1.0] | 0.9 [0.8, 0.98] |
| Subcutaneous Emphysema | **0.97** [0.9, 1.0] | 0.0 [0.0, 0.0] | **0.8** [0.68, 0.92] | 0.0 [0.0, 0.0] | **0.88** [0.79, 0.95] | 0.0 [0.0, 0.0] | **0.87** [0.78, 0.94] | 0.0 [0.0, 0.0] |
| Tortuous Aorta | **0.83** [0.69, 0.95] | 0.79 [0.64, 0.91] | 0.88 [0.75, 0.97] | **0.94** [0.84, 1.0] | 0.85 [0.75, 0.93] | **0.86** [0.75, 0.94] | 0.84 [0.73, 0.93] | **0.85** [0.74, 0.93] |
| **Macro-averages** | | | | | | | | |
| CheXpert-5 (CXP-5) | **0.75** [0.71, 0.78] | 0.72 [0.69, 0.76] | 0.82 [0.79, 0.85] | **0.87** [0.84, 0.9] | **0.78** [0.75, 0.8] | 0.78 [0.75, 0.81] | 0.67 [0.64, 0.71] | **0.69** [0.65, 0.72] |
| CheXpert-7 (CXP-7) | **0.71** [0.66, 0.75] | 0.65 [0.62, 0.68] | 0.77 [0.73, 0.81] | **0.87** [0.83, 0.9] | **0.73** [0.7, 0.77] | 0.71 [0.69, 0.74] | **0.65** [0.61, 0.69] | **0.65** [0.61, 0.68] |
| CheXpert-13 (CXP-13) | **0.66** [0.63, 0.69] | 0.63 [0.61, 0.65] | 0.68 [0.63, 0.73] | **0.81** [0.75, 0.85] | **0.63** [0.6, 0.65] | 0.62 [0.6, 0.64] | **0.57** [0.54, 0.6] | 0.56 [0.54, 0.59] |
| LT-only | **0.73** [0.69, 0.77] | 0.53 [0.48, 0.57] | **0.75** [0.7, 0.8] | 0.73 [0.69, 0.77] | **0.72** [0.68, 0.75] | 0.59 [0.55, 0.62] | **0.71** [0.67, 0.74] | 0.59 [0.55, 0.63] |
| CXR-LT | **0.7** [0.67, 0.72] | 0.58 [0.56, 0.6] | 0.71 [0.68, 0.75] | **0.77** [0.74, 0.8] | **0.67** [0.65, 0.69] | 0.61 [0.58, 0.62] | **0.64** [0.61, 0.66] | 0.58 [0.55, 0.6] |
| Micro | **0.69** [0.67, 0.71] | 0.62 [0.6, 0.64] | 0.76 [0.74, 0.78] | **0.8** [0.78, 0.82] | **0.72** [0.71, 0.74] | 0.7 [0.68, 0.72] | **0.67** [0.65, 0.69] | 0.64 [0.62, 0.66] |

 ## C.3 Scene Graph Evaluation: Finding Boxes

Table 6: Evaluation of finding bounding boxes against 6 finding classes from MS-CXR (see Sec. 4.1). We show finding-level scores, macro-averages over different subsets and the micro-average, with 95% confidence intervals (bootstrapping, $n = 1000$). We excluded 2 of the 8 finding classes, because there are no samples that have positive annotations from MS-CXR, Chest ImaGenome and our dataset.

| | MS-CXR [31] | | | | | |
| --- | --- | --- | --- | --- | --- | --- |
| | [IoU@30] | | [IoP@30] | | [IoT@30] | |
| | Ours | Chest ImaG. | Ours | Chest ImaG. | Ours | Chest ImaG. |
| Atelectasis | **0.28** [0.12, 0.42] | 0.1 [0.02, 0.2] | **0.5** [0.34, 0.66] | 0.14 [0.05, 0.28] | 0.83 [0.71, 0.93] | **0.85** [0.74, 0.94] |
| Cardiomegaly | 0.96 [0.93, 0.98] | **0.97** [0.95, 0.99] | **1.0** [1.0, 1.0] | **1.0** [1.0, 1.0] | 0.96 [0.93, 0.98] | **0.99** [0.98, 1.0] |
| Consolidation | **0.31** [0.19, 0.45] | 0.2 [0.1, 0.31] | **0.41** [0.29, 0.54] | 0.24 [0.12, 0.35] | 0.91 [0.81, 0.98] | **0.98** [0.93, 1.0] |
| Edema | **0.52** [0.32, 0.71] | **0.52** [0.32, 0.71] | **0.52** [0.32, 0.71] | **0.52** [0.32, 0.71] | **1.0** [1.0, 1.0] | **1.0** [1.0, 1.0] |
| Pneumonia | **0.48** [0.41, 0.57] | 0.28 [0.21, 0.35] | **0.58** [0.5, 0.66] | 0.34 [0.26, 0.42] | 0.93 [0.88, 0.97] | **1.0** [1.0, 1.0] |
| Pneumothorax | 0.14 [0.1, 0.18] | **0.15** [0.1, 0.2] | 0.14 [0.1, 0.19] | **0.15** [0.11, 0.2] | 0.96 [0.93, 0.98] | **0.98** [0.96, 1.0] |
| Macro | **0.45** [0.4, 0.5] | 0.37 [0.33, 0.41] | **0.53** [0.48, 0.58] | 0.4 [0.36, 0.44] | 0.93 [0.9, 0.95] | **0.97** [0.95, 0.98] |
| Micro | **0.51** [0.47, 0.54] | 0.45 [0.42, 0.49] | **0.56** [0.52, 0.6] | 0.48 [0.45, 0.52] | 0.94 [0.92, 0.96] | **0.98** [0.97, 0.99] |

Table 7: Evaluation of finding bounding boxes against 18 finding classes from REFLACX (see Sec. 4.1). We show finding-level scores, macro-averages over different subsets and the micro-average, with 95% confidence intervals (bootstrapping, $n = 1000$). Note that we excluded 11 of the 29 finding classes, because there are no samples that have positive annotations from REFLACX, Chest ImaGenome and our dataset.

| | REFLACX [32] all phases | | | | | |
| --- | --- | --- | --- | --- | --- | --- |
| | [IoU@30] | | [IoP@30] | | [IoT@30] | |
| | Ours | Chest ImaG. | Ours | Chest ImaG. | Ours | Chest ImaG. |
| Abnormal mediastinal contour | 0.08 [0.0, 0.31] | **0.25** [0.0, 0.57] | 0.08 [0.0, 0.31] | **0.25** [0.0, 0.57] | **1.0** [1.0, 1.0] | **1.0** [1.0, 1.0] |
| Acute fracture | **0.0** [0.0, 0.0] | **0.0** [0.0, 0.0] | **0.0** [0.0, 0.0] | **0.0** [0.0, 0.0] | **1.0** [1.0, 1.0] | 0.0 [0.0, 0.0] |
| Atelectasis | **0.29** [0.26, 0.33] | 0.15 [0.12, 0.17] | **0.47** [0.44, 0.51] | 0.2 [0.17, 0.23] | 0.76 [0.73, 0.78] | **0.93** [0.91, 0.94] |
| Consolidation | **0.39** [0.33, 0.45] | 0.27 [0.22, 0.32] | **0.51** [0.45, 0.57] | 0.34 [0.28, 0.4] | 0.8 [0.74, 0.85] | **0.95** [0.92, 0.97] |
| Emphysema | **1.0** [1.0, 1.0] | **1.0** [1.0, 1.0] | **1.0** [1.0, 1.0] | **1.0** [1.0, 1.0] | **1.0** [1.0, 1.0] | **1.0** [1.0, 1.0] |
| Enlarged cardiac silhouette | **0.96** [0.94, 0.97] | **0.96** [0.95, 0.97] | **1.0** [0.99, 1.0] | 0.99 [0.99, 1.0] | 0.96 [0.95, 0.97] | **0.98** [0.97, 0.99] |
| Enlarged hilum | **0.5** [0.0, 1.0] | **0.5** [0.0, 1.0] | 0.5 [0.0, 1.0] | **0.8** [0.23, 1.0] | **0.5** [0.0, 1.0] | **0.5** [0.0, 1.0] |
| Fracture | **0.0** [0.0, 0.0] | **0.0** [0.0, 0.0] | **0.0** [0.0, 0.0] | **0.0** [0.0, 0.0] | **0.0** [0.0, 0.0] | **0.0** [0.0, 0.0] |
| Groundglass opacity | 0.28 [0.22, 0.34] | **0.31** [0.25, 0.38] | **0.48** [0.4, 0.54] | 0.38 [0.32, 0.45] | 0.77 [0.71, 0.83] | **0.96** [0.93, 0.99] |
| Hiatal hernia | 0.19 [0.0, 0.43] | **0.4** [0.17, 0.67] | 0.27 [0.06, 0.5] | **0.4** [0.17, 0.67] | 0.94 [0.78, 1.0] | **1.0** [1.0, 1.0] |
| High lung volume / emphysema | 0.48 [0.25, 0.7] | **0.58** [0.35, 0.79] | **0.58** [0.35, 0.79] | **0.58** [0.35, 0.79] | 0.9 [0.75, 1.0] | **1.0** [1.0, 1.0] |
| Interstitial lung disease | 0.5 [0.0, 1.0] | **0.8** [0.0, 1.0] | **0.8** [0.0, 1.0] | **0.8** [0.0, 1.0] | 0.8 [0.12, 1.0] | **1.0** [1.0, 1.0] |
| Lung nodule or mass | **0.18** [0.08, 0.31] | 0.09 [0.02, 0.2] | **0.21** [0.1, 0.35] | 0.09 [0.02, 0.2] | 0.89 [0.77, 0.97] | **0.91** [0.8, 0.98] |
| Pleural abnormality | **0.16** [0.13, 0.19] | 0.14 [0.11, 0.17] | **0.2** [0.17, 0.23] | 0.17 [0.14, 0.2] | 0.91 [0.88, 0.93] | **0.92** [0.9, 0.94] |
| Pleural effusion | **0.42** [0.2, 0.65] | 0.37 [0.17, 0.6] | **0.53** [0.3, 0.75] | **0.53** [0.31, 0.75] | **1.0** [1.0, 1.0] | 0.95 [0.82, 1.0] |
| Pleural thickening | **0.0** [0.0, 0.0] | **0.0** [0.0, 0.0] | **0.0** [0.0, 0.0] | **0.0** [0.0, 0.0] | **1.0** [1.0, 1.0] | **1.0** [1.0, 1.0] |
| Pneumothorax | 0.04 [0.01, 0.08] | **0.13** [0.07, 0.21] | 0.04 [0.01, 0.08] | **0.13** [0.07, 0.21] | 0.9 [0.84, 0.96] | **0.96** [0.92, 0.99] |
| Pulmonary edema | 0.51 [0.45, 0.56] | **0.58** [0.53, 0.63] | 0.55 [0.5, 0.6] | **0.58** [0.53, 0.63] | 0.95 [0.93, 0.97] | **1.0** [1.0, 1.0] |
| Macro | 0.34 [0.27, 0.42] | **0.37** [0.3, 0.45] | **0.41** [0.34, 0.49] | **0.41** [0.33, 0.49] | 0.84 [0.78, 0.91] | **0.87** [0.8, 0.95] |
| Micro | **0.45** [0.44, 0.47] | 0.42 [0.4, 0.43] | **0.54** [0.53, 0.56] | 0.46 [0.44, 0.47] | 0.87 [0.86, 0.88] | **0.95** [0.94, 0.96] |

# D  Dataset Structure

## D.1  Scene Graph Structure

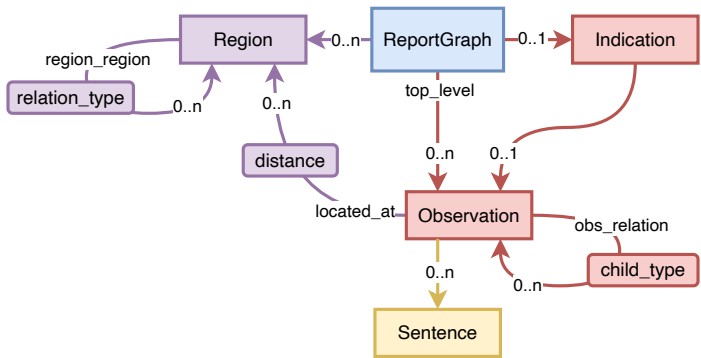

Figure 13: Scene graph structure overview.

**Sentence Nodes**  Sentence nodes are directly associated with raw sentences in the report, i.e. there is exactly one sentence node per identified report sentence. They contain the following attributes:

- `sent_id`: Identifier, unique per study.
  Example: S01.

- `section`: Name of the section that the sentence belongs to, as specified in the report. If the sentence is not part of a section, `FINAL_REPORT_NO_SECTION` or `PRE_FINAL_REPORT_NO_SECTION` are used.
  Examples: FINDINGS, IMPRESSION, REASON_FOR_EXAM.

- `section_type`: The identified type of section used for classifying the type of content of the sentence. `IGNORE` is used for irrelevant sections.
  Examples: FINDINGS, IMPRESSION, INDICATION.

- `sentence`: The raw sentence as written in the report.

**Observation Nodes**  Observation nodes are created for each individually described aspect (i.e. observation) in the report's `FINDINGS` or `IMPRESSION` section. Hereby, a single sentence may be related to several observation nodes and a single observation may be derived from several sentences (if they describe related aspects). Observation nodes are structured hierarchically, i.e. they may have other observation nodes as parents. An observation node contains the following attributes:

- `obs_id`: Identifier, unique per study.
  Example: O01.
  For child nodes this also contains the parent id, e.g. O01.02.

- `summary_sentence`: Textual description of the observation, directly derived from the associated report sentences. In some cases, this may be an exact copy of the report sentences but it may also paraphrase parts of it.

- `name`: Abbreviated version of the `summary_sentence`.

- `child_level`: Hierarchy level, 0 for top-level, larger numbers for deeper hierarchy levels.

- `child_type`: Type of parent-child relation.
  Possible options: `regional_distinction`, `related_region`, `associated_with`, `device_part`, `recommendation`, `comparison_only`.

- `regions`: List of associated regions, each paired with an optional list of distance annotations.
  Example: [("heart", ["1 cm above"])]

- `non_resolved_regions`: Similar to `regions` but with regions that could not be semantically mapped to reference definitions.

- **laterality**: Laterality of the region.
  Possible options: `left`, `right`, `likely bilateral`, `bilateral`, `unknown`.
- **default_regions**: List of regions that have been added because they are defaults for the identified findings (`obs_entities`).
- **obs_entities**: List of (directly) associated findings.
  Example: `["pleural effusion"]`.
- **obs_entities_parents**: List of findings that are considered parents of findings in `obs_entities`.
- **non_resolved_obs_entities**: Similar to `obs_entities` but with findings that could not be semantically mapped to reference definitions.
- **obs_categories**: List of associated finding super-categories.
  Example: `["ANATOMICAL_FINDING"]`.
- **obs_subcategories**: List of associated finding sub-categories.
  Example: `["LUNG_FIELD"]`.
- **probability**: Likelihood of the observation being positive. Short term, derived from what is mentioned in the report.
- **certainty**: How certain is the observation. Derived from `probability`.
  Possible options: `certain`, `likely`, `uncertain`, `comparison_only`, `recommendation`.
- **positiveness**: Whether the observation is positive or negative. Derived from `probability`.
  Possible options: `pos`, `neg`, `comparison_only`, `recommendation`.
- **modifiers**: Modifiers of the finding. Dictionary with keys for each type of modifier and lists of the individual modifier values.
  Possible modifier type: `severity`, `texture`, `spread`, `temporal`.
  Example: `{"severity": ["mild"], "spread": ["focal"]}`.
- **change_sentence**: Optional textual description of any changes to the prior study of the same patient, if it was mentioned in the report.
- **changes**: List of change types mentioned in the `change_sentence`.
  Example: `worsening`.
- **from_report**: Whether this observation was explicitly mentioned in the report (`true`) or automatically added (`false`).
- **obs_quality**: Extraction quality of the observation, consisting of several individual aspects. See Tab. 8.
- **localization**: Bounding boxes for this observation, for each associated image. Dictionary with keys equaling image ids (each study may correspond to several images). Values contain:
  - **image_id**
  - **bboxes**: List of bounding boxes in the $(x_1, y_1, x_2, y_2)$ format in original image-pixel coordinates.
  - **localization_reference_ids**: List of region names from which the bounding boxes are derived.
  - **missing_localization**: List of associated region names for which no localization is available for this image.
  - **is_fallback**: Whether this localization is a fallback, i.e. the original region localization was not available but a more coarse localization was used instead.
  - **localization_quality**: Quality of the localization. See Tab. 8.

**Region Nodes**  Region nodes are created for each anatomical structure mentioned in any observation and for key regions. They contain the following attributes:

- **region**: Name of the region and unique identifier within each study.
  Example: `left lung`.
- **laterality**: Laterality of the region.
  Possible options: `left`, `right`, `bilateral`, `unknown` (i.e. not clearly definable).

- **localization**: Bounding boxes for this region, for each associated image. Same format as for observation nodes.
- **region_localization_quality**: Quality of the localization attribute. See Tab. 8.

**Indication Node**    Each study contains an optional indication node with information extracted from the INDICATION section. If present, it contains the following attributes:

- **indication_summary**: Summary of the indication, directly derived from the INDICATION section of the report, but typically paraphrased.
- **patient_info**: Any information about the patient, if mentioned in the INDICATION section. A subset of the content in indication_summary.
- **indication**: Indication for the study, if mentioned in the INDICATION section. A subset of the content in indication_summary.
- **evaluation**: Any required evaluation of the patient (i.e. what should be evaluated with this study), if mentioned in the INDICATION section. A subset of the content in indication_summary.
- **associated_sentence_ids**: List of sent_ids from sentence nodes that are related to the indication.
- **associated_obs_ids**: List of obs_ids from observation nodes that are related to the indication.
- **answer_for_indication**: A single observation node containing the answer to the question (implicitly) asked by the provided indication. This is a special observation node with obs_id = OIND. Its textual description is directly derived from the FINDINGS and IMPRESSION sections but conditioned on the INDICATION section.

**Root Node and Relations**    Each study contains a single root node called the ReportGraph. It contains general metadata about the study and its scene graphs:

- **patient_id**: Unique patient ID, the subject_id from MIMIC-CXR.
- **study_id**: Unique study ID, from MIMIC-CXR. Each patient may have several studies.
- **study_quality**: The overall extraction quality of the scene graph for this study, consisting of several individual aspects. See Tab. 8.
- **study_img_localization_quality**: Dictionary of localization qualities for each image with keys corresponding to image IDs. See Tab. 8.

Additionally, it is connected to all other nodes and links to the top-level (root) observations. Thus, it contains the following:

- **sentences**: List of all sentence nodes.
- **observations**: Dictionary of all observation nodes, indexed by their obs_id.
- **top_level_obs_ids**: List of all top-level (root) observation node IDs, i.e. their obs_ids.
- **regions**: Dictionary of all region nodes, indexed by their region attribute.
- **indication**: The indication node, if it exists.

Nodes can also be connected by the following relations:

- **located_at_relations** (observation↔region): Specifies where an observation is located with the following additional attributes:
  - **distances**: List of distance annotations, e.g. ["3cm above"].
  - **where_specified**: How this relation was derived. Possible options: direct, bilateral, sub_region.
- **obs_relations** (observation↔observation): Specifies a parent-child relation between two observations, with the following additional attribute:

- **child_type**: Type of parent-child relation.
  Possible options: `regional_distinction`, `related_region`, `associated_with`, `device_part`, `recommendation`, `comparison_only`.

- **obs_sent_relations** (observation↔sentence): Specifies from which sentences an observation was derived.

- **region_region_relations** (region↔region): Specifies a relation between two regions with the following additional attribute:

  - **relation_type**: Type of relation.
    Possible options: `sub_region`, `bilateral` (the bilateral version of a region), `left` (the left version of a region), `right` (the right version of a region).

## D.2 Question-Answer Structure

**QA-Pair**  Each question-answer pair consists of a free-text question (attribute `question`), an answer consisting of structured answer parts (attribute `answers`). Additionally, it contains the following metadata:

- **question_id**: Identifier, unique within the associated study.

- **question_type**: The QA-template used to generate this QA-pair.

- **question_strategy**: The strategy used to generate QA-pair. See Sec. 3.2 and Appendix E.2.2.

- **variables**: Key-value pairs of variables (and their values) used during generation, e.g. to fill the template. See Appendix E.2.1.

- **obs_ids**: List of `obs_idss` of observation nodes (in the scene graph) from which the answer is derived.

- **contains_report_answers**: Whether any of the answer parts was derived from the report, i.e. from observation nodes.

- **contains_template_answers**: Whether any of the answer parts was generated based on a template.

- **extraction_quality**: The overall extraction quality of the associated observations in the scene graph, consisting of several individual aspects. See Tab. 8.

- **question_img_localization_quality**: Quality of the localizations per image. See Tab. 8.

- **question_quality**: The overall question-answer text quality, consisting of several individual aspects. See Tab. 9.

- **rating**: The overall rating of the QA-pair. See Appendix D.3

Answers are structured hierarchically, consisting of a list of answer parts (attribute `answers`) and sub-answers (children) of these answers, where there can be several hierarchy levels. The hierarchy levels are derived from the parent-child structure of associated observation nodes (based on `obs_relations`, Appendix D.1). Additionally, there are different types of answer parts:

- **main_answer**: Required to answer the question. There is always at least one main-answer per question.

- **details**: Providing additional details for the main answer, which are however not mandatory to answer the question.

- **related_information**: Not directly answering the question, but may be related and provides context.

Each individual answer part contains the following attributes:

- **answer_id**: Identifier, unique within each study. Contains the `question_id`.

- **text**: The answer text. Either generated from a template or based on `summary_sentence` in the observation node (Appendix D.1).

- **answer_level**: Hierarchy level, 0 for top-level answer part, larger numbers for deeper hierarchy levels (sub-answers).

- **answer_type**: Type of answer part.
  Possible options: `main_answer`, `details`, `related_information`.

- **name_tag**: Abbreviated version of the `text`. Either generated from a template or based on `name` in the observation node (Appendix D.1).

- **laterality**: Laterality of the region. See observation node (Appendix D.1).
  Possible options: `left`, `right`, `likely bilateral`, `bilateral`, `unknown`.

- **regions**: List of associated regions. See observation node (Appendix D.1). Distances are not provided here.
  Example: `["heart"]`

- **obs_entities**: List of (directly) associated findings. See observation node (Appendix D.1).
  Example: `["pleural effusion"]`.

- **obs_entities_parents**: List of findings that are considered parents of findings in `obs_entities`. See observation node (Appendix D.1).

- **obs_categories**: List of associated finding super-categories. See observation node (Appendix D.1).
  Example: `["ANATOMICAL_FINDING"]`.

- **obs_subcategories**: List of associated finding sub-categories. See observation node (Appendix D.1).
  Example: `["LUNG_FIELD"]`.

- **certainty**: How certain is the observation. See observation node (Appendix D.1).
  Possible options: `certain`, `likely`, `uncertain`, `comparison_only`, `recommendation`.

- **positiveness**: Whether the observation is positive or negative. See observation node (Appendix D.1).
  Possible options: `pos`, `neg`, `comparison_only`, `recommendation`.

- **modifiers**: Modifiers of the finding. List of pairs of modifier type and value. See observation node (Appendix D.1).
  Possible modifier type: `severity`, `texture`, `spread`, `temporal`.
  Example: `[("severity", "mild"), ("spread", "focal")]`.

- **localization**: Bounding boxes for this answer part, for each associated image. Dictionary with keys equaling image ids (each study may correspond to several images). See observation node (Appendix D.1).

- **sub_answers**: List of child answers (deeper in the hierarchy). Each sub-answer is another answer-part with all attributes and potentially further sub-answers.

- **from_report**: Whether this answer part is derived from the report, i.e. an observation node (`true`), or from a template (`false`).

- **extraction_quality**: The overall extraction quality of the associated observations in the scene graph, consisting of several individual aspects. See Tab. 8.

- **answer_quality**: The overall answer text quality, consisting of several individual aspects. See Tab. 9.

## D.3 Quality

**Ratings** We distinguish between the following overall ratings for each QA-pair:

- **A++**: Perfect and complete content; all information in the answer is explicitly mentioned in the report.

- **A+**: Perfect and mostly complete content; all information in the answer is explicitly mentioned in the report, but some minor details may be missing or irrelevant.

- **A**: Very good content with minor issues not affecting the overall quality; some tags or boxes may be inferred or minor issues (e.g. grammatical) may be present in the text.

- **B**: Good content; factually correct answers, which may however be not fully complete or slightly unclear.

- **C**: Poor content; answers may be misleading or contain completely unclear information.

- **D**: Incorrect content; answers may be contradicting the report and are not usable.

- **not rated**: Quality could not be assessed, e.g. due to invalid LLM-rater outputs.

These ratings are derived based on individual aspects that will be described in the following paragraphs. Possible quality levels for each aspects and the resulting rating are presented in Tabs. 8 and 9. The final rating is computed as the minimum (worst) rating over all individual aspects.

**Scene Graph Extraction Quality**    For each scene graph, we provide a quality rating based on how well it could be constructed/extracted. Tab. 8 shows the considered aspects with their potential quality levels and resulting ratings.

Table 8: Quality levels for the 6 scene graph quality aspects, with their resulting ratings.

| | Quality level | Value | Resulting rating |
|---|---|---|---|
| **How well are region tags identified?** (attribute `regions`) | | | |
| Region extraction | NO_REGIONS | 0 | B |
| | DEFAULT_REGIONS_ONLY | 1 | B |
| | CONTAINS_DEFAULT_REGIONS | 2 | A |
| | CONTAINS_NON_RESOLVED_REGIONS | 3 | A |
| | RESOLVED_REGIONS_ONLY | 4 | A++ |
| **How well are finding tags identified?** (attribute `obs_entities`) | | | |
| Finding extraction | NO_ENTITIES | 0 | B |
| | CONTAINS_NON_RESOLVED_ENTITIES | 1 | A |
| | RESOLVED_ENTITIES_ONLY | 2 | A++ |
| **How well are textual descriptions extracted?** (attributes `summary_sentence` and `name`) | | | |
| Description extraction | CHANGE_IN_SENTENCE_OR_NAME | 0 | B |
| | UNDERSCORES_IN_SENTENCE_OR_NAME | 1 | A |
| | NO_ISSUES | 2 | A++ |
| **How well are mentions of change extracted?** (attributes `change_sentence` and `change`) | | | |
| Change extraction | CHANGE_SENTENCE_REMOVED | 0 | B |
| | UNDERSCORES_IN_CHANGE_SENTENCE | 1 | A |
| | CONTAINS_NON_RESOLVED_CHANGES | 2 | A |
| | NO_ISSUES | 3 | A++ |
| **Have there been any issues in the extraction and scene graph construction pipeline?** | | | |
| Extraction issues | DISCARDED | -1 | D |
| | NON_INTERPRETABLE | 0 | C |
| | MOSTLY_INTERPRETABLE | 1 | B |
| | IGNORABLE | 2 | A |
| | FIXABLE | 3 | A+ |
| | NO_ISSUES | 4 | A++ |
| **How well could observations/regions be localized?** (attribute `localization`)) | | | |
| Localization | NO_LOCALIZATION | 0 | B |
| | FALLBACK_LOCALIZATION | 1 | B |
| | INCOMPLETE_LOCALIZATION | 2 | A |
| | BBOX_LOCALIZATION | 3 | A++ |
| | BBOX_AND_MASK_LOCALIZATION | 4 | A++ |

Finding extraction is also referred to as entity extraction, description extraction as sentence/name quality.

**QA Text Quality**   For each QA-pair, we provide quality rating for its text, i.e. the question text and the textual descriptions in its answer parts. Tab. 9 shows the considered aspects with their potential quality levels and resulting ratings.

Table 9: Quality levels for the 5 QA-pair text quality aspects, with their resulting ratings.

| | Quality level | Value | Resulting rating |
|---|---|---|---|
| **Entailment** | **Does the answer factually align with the original report?** | | |
| | (rated per answer-part, given the question and the report) | | |
| | NON_ALIGNED_CONTRADICTING | -3 | D |
| | NON_ALIGNED_MISLEADING | -2 | C |
| | NON_ALIGNED_NON_INFERABLE | -1 | B |
| | ALIGNED_GENERAL_STATEMENT | 0 | A |
| | ALIGNED_NEGATIVE_NOT_MENTIONED | 1 | A+ |
| | ALIGNED_INFERABLE | 2 | A++ |
| | ALIGNED_MENTIONED | 3 | A++ |
| **Relevance** | **Is the answer relevant for the given question?** | | |
| | (rated per answer-part, given the question but independent of the report) | | |
| | IRRELEVANT_INFO | -2 | A |
| | REDUNDANT_INFO | -1 | A |
| | RELATED_INFO | 0 | A+ (A++ for related_information answer) |
| | RELEVANT_MAIN_ANSWER | 1 | A++ (A for related_information answer) |
| **Completeness** | **Does the answer cover all aspects in the report that are relevant to the question?** | | |
| | (rated for the full answer, given the question and the report) | | |
| | INCOMPLETE_MISLEADING | -2 | C |
| | INCOMPLETE_NON_MISLEADING | -1 | B |
| | NOT_ANSWERED | 0 | B |
| | DETAILS_MISSING | 1 | A+ |
| | FULLY_COMPLETE | 2 | A++ |
| **Question clarity** | **Is the generated question clear and grammatically correct?** | | |
| | (rated for the question, given nothing else) | | |
| | UNANSWERABLE | -3 | C |
| | UNRELATED_TO_CHEST_XRAY | -2 | B |
| | UNCLEAR_QUESTION | -1 | B |
| | GRAMMATICAL_ERRORS | 0 | A |
| | UNUSUAL_SENTENCE_STRUCTURE | 1 | A |
| | OPTIMAL | 2 | A++ |
| **Answer clarity** | **Is the answer clear and grammatically correct?** | | |
| | (rated per answer-part, given nothing else) | | |
| | NOT_UNDERSTANDABLE | -2 | C |
| | UNCLEAR_ANSWER | -1 | B |
| | GRAMMATICAL_ERRORS | 0 | A |
| | UNUSUAL_SENTENCE_STRUCTURE | 1 | A |
| | OPTIMAL | 2 | A++ |

# E   Dataset Construction Details

## E.1   Scene Graph Construction

### E.1.1   Region Localization

We use the CXAS [22], [23] model to predict segmentation masks of 158 anatomical structures on the 377,110 CXRs from MIMIC-CXR-JPG [15], [24], [25]. Additionally, we use the bounding boxes provided by the Chest ImaGenome [15], [17], [19] dataset, which are provided for 29 anatomical structures in most frontal images of MIMIC-CXR. The masks predicted by CXAS are post-processed with morphological operations to filter out outlier pixels.

We specify 257 localized regions in our reference definitions. For each of these regions, we define how the bounding boxes are derived. We consider the following options:

- **CXAS masks**: Some regions are directly associated with one of the 158 anatomical structures for which the CXAS model predicts segmentation masks. In these cases, we compute the bounding box around the predicted segmentation mask.

- **Chest ImaGenome boxes**: Some regions are directly associated with one of the 29 anatomical structures for which Chest ImaGenome provides bounding boxes. In such cases, we use these provided bounding boxes if no CXAS masks are associated.

- **Bilateral regions**: Some regions refer to a pair of bilateral regions (e.g. *lungs* refers to *left lung* and *right lung*). In these cases, we simply use the two bounding boxes of the left and right versions, but do not fuse them.

- **Parent regions**: For some regions we do not have exact correspondences to available masks or boxes but we have available sub-regions. In these cases, we compute the super bounding box, a single box, around all specified child regions.

- **Fusions**: In some rare cases, we combine multiple individual masks or bounding boxes. We compute intersections or unions of boxes or masks, before inferring the final bounding box.

After computing all regions, we filter out regions with a too small bounding box area. For images where a specific region is not available, we try to use alternative regions as fallbacks instead, e.g. using a more coarse parent regions as an alternative. Note that this is often the case for lateral images as there no Chest ImaGenome boxes are available.

### E.1.2   Information Extraction

**Extracting the Sentences**   First, we extract individual sentences from the reports, detect their sections (e.g. FINDINGS, IMPRESSION, INDICATION, ...), discard sentences without relevant information, and merge sentences containing similar information (e.g. if findings are described in both the FINDINGS and IMPRESSION section). Therefore, each full report is passed in a single step to the LLM, which predicts the individually separated sentences as well as their sections and related sentences. We use the prompt shown in Listing 1 (with few-shot examples similar to Listing 2) and apply it to the full radiology report. After parsing the LLM outputs, we apply the Stanza [39] tokenizer to each identified sentence and try to further split it. The LLM also identified potentially related sentences. We use this information to identify sentence clusters containing related information. Such sentence clusters are the basis for the next step, i.e. observation extraction. We successfully extracted sentence from 227 626 studies (reports) while having parse errors for 209 studies.

Listing 1: LLM prompt used for sentence extraction.

```
747
748  Extract all sentences from the given textual report.
749  You will be given a (free-text) medical radiology report describing
750      ↪ one or more chest X-rays of a single patient.
751
752  # Rules:
753  - Split the report into sentences and extract all sentences in the
754      ↪ report.
755  - Do not rewrite the sentences!
756  - For each sentences, identify the its section name (written in the
757      ↪ report).
758  If a sentence is not part of a section but is part of the "FINAL
759      ↪ REPORT", then use "FINAL\_REPORT\_NO\_SECTION". If a sentence
760      ↪ is not part of a section but the sentence is before the "FINAL
761      ↪ REPORT", use the section name "PRE\_FINAL\_REPORT\_NO\_SECTION"
762      ↪ .
763  - For each sentence, classify the content written therein into one of
764      ↪ the following types: [EXAM\_TECHNIQUE, INDICATION, FINDINGS,
765      ↪ IMPRESSION, PRE\_FINAL\_REPORT, IGNORE]. This is typically
766      ↪ inferred from the section name but may also be influenced by
767      ↪ the content of the sentence. Some example sections names for
768      ↪ each type are given below:
769      EXAM\_TECHNIQUE: EXAMINATION, EXAM, TECHNIQUE
770      INDICATION: INDICATION, INDICATIONS, HISTORY, CLINICAL HISTORY,
771          ↪ CLINICAL, REASON, REASON FOR EXAM
772      FINDINGS: FINDING, FINDINGS
773      IMPRESSION: IMPRESSION, IMPRESSIONS, RECOMMENDATION
774      PRE\_FINAL\_REPORT: WET\_READ, WET\_READ\_VERSION\_#1, PRE\_FINAL\
775          ↪ _REPORT\_NO\_SECTION
776      IGNORE: COMPARISON, COMPARISONS, REFERENCE EXAM, NOTIFICATION
777  - Split the report into individual sentences and report each sentence
778      ↪ in its own line, removing any newlines present in the sentence.
779  - For enumerations: each point is considered an independent sentence!
780      ↪ Remove the numbering.
781  - Specify sentence IDs of similar, previous sentences that each
782      ↪ sentence could be merged with. A sentence should be merged with
783      ↪  all previous sentences that either describe the same aspect or
784      ↪  that refer to each other (e.g. if a sentence provides further
785      ↪ details to a previous one). A bullet point may also be
786      ↪ associated with a sentence, even if the other sentence has a
787      ↪ different bullet number or none at all.
788  - Follow the examples given below!
789
790  # Examples:
791   <FEWSHOT>
792
793  # Input Report (extract data from this report):
794  --- START OF REPORT ---
795   <REPORT>
796  --- END OF REPORT ---
797
798  # Hints:
799  - Infer the output format from the examples!
800  - Do not add any explanations or text BEFORE or AFTER the extracted
801      ↪ sentences, i.e. start with the first sentence!
802
803  # Proceed:
804
```

Listing 2: Few-shot example for sentence extraction.

```
***Example: Report***
--- START OF REPORT ---
                              FINAL REPORT
PORTABLE CHEST OF \_\_\_

COMPARISON:  \_\_\_ radiograph.

FINDINGS:  No pleural effusion or pneumothorax.
--- END OF REPORT ---

***Example 5: Output***
[S01] FINAL\_REPORT\_NO\_SECTION(EXAM\_TECHNIQUE) - merge with []:
    ↪ PORTABLE CHEST OF \_\_\_
[S02] COMPARISON(IGNORE) - merge with []: \_\_\_ radiograph.
[S03] FINDINGS(FINDINGS) - merge with []: No pleural effusion or
    ↪ pneumothorax.
```

**Extracting the Observations** In this step, we consider each sentence cluster (as identified during sentence extraction), in the FINDINGS and IMPRESSION sections. A sentence cluster contains one or more sentences that describe related aspects and may stretch over one of both of these sections. From each of these clusters, we now extract mentioned observations using the prompt shown in Listing 3 with few-shot examples similar to Listing 4. We apply this prompt to each sentence cluster individually and extract zero, one, or multiple observations each. The output is provided in the json-format and follows a similar structure as the final observation node, but we optimized it to be easy to fill by the LLM. The LLM is allowed to freely assign values to each of the json-fields. For `name` and `summary_sentence`, we prompt the model to stay close to the original sentence, but it must remove any mentions of change and only keep the part relevant to the individual observation (if several observations are mentioned in one sentence). We successfully extracted observations from 227 266 studies (reports) while having parse errors for 360 studies.

Listing 3: LLM prompt used for observation extraction.

```
Extract structured information from the given textual report.
You will be given sentences from a (free-text) medical radiology
    ↪ report describing one or more chest X-rays of a single patient.

# Guidelines:
<GUIDE>

# Rules:
- Follow the examples given below!

# Examples:
<FEWSHOT>

# Hints:
- Check for any "change" modifiers (see guidelines).
- If there is a "change" modifier, rewrite the "summary\_sentence"
    ↪ such that it describes only what is visible in the current
    ↪ image, without any mentions of change or comparisons! Describe
    ↪ the change in the "change\_sentence". Do this for all top-level
    ↪  AND child observations.
- Make sure to include all children of observations, even if they
    ↪ repeat information from the parent!

# Proceed with the Input Sentence:
Sentence(s): <SENT>
Output JSON-List:
```

Listing 4: Few-shot example for observation extraction.

```
Sentence(s): Left more than right basilar atelectasis.
Output JSON-List:
[
    {
        "name": "bibasilar atelectasis", "entity": "atelectasis",
        "probability": "positive", "change": null,
        "summary\_sentence": "Bibasilar atelectasis.",
        "change\_sentence": null,
        "regions": ["bibasilar"],
        "children": [
            {
                "child\_type": "regional\_distinction", "name": "left
                    ↪ basilar atelectasis", "entity": "atelectasis",
                "probability": "positive", "change": null,
                "summary\_sentence": "Left more than right basilar
                    ↪ atelectasis.",
                "change\_sentence": null,
                "regions": ["left basilar"]
            }
        ]
    }
]
```

**Extracting the Indication** Next, we extract information about the INDICATION section and detect which FINDINGS or IMPRESSION sentences may provide information related to the indication. Therefore, the extracted INDICATION sentences and a list of all FINDINGS and IMPRESSION sentences are passed to the LLM using the prompt shown in Listing 5 with few-shot examples similar to Listing 6. The LLM predicts a json-structure containing several text fields for summaries of aspects in the indication, an `answer_for_indication` derived from the FINDINGS and IMPRESSION section, as well as relevant sentence IDs. We successfully extracted indictions from 227 596 studies (reports) while having parse errors for 30 studies.

Listing 5: LLM prompt used for indication extraction.

```
Extract structured information from the given (free-text) medical
    ↪ report.
You will be given the indication sentence from a report and
    ↪ additionally the sentences from the findings section.

# Rules:
- Extract / summarize the given indication information. Use only the
    ↪ provided indication sentence.
- Additionally, identify the finding sentences associated with the
    ↪ indication, i.e. the sentence that answer the quesiton of the
    ↪ indication or are highly relevant to it. Based on these finding
    ↪  sentences, provide an answer to the question asked in the
    ↪ indication.
- Follow the examples given below!

# Examples:
 <FEWSHOT>

# Hints:
- For each attribute, write full sentences instead of single terms or
    ↪ bullet points.
- In the "answer\_for\_indication", describe in YOUR OWN WORDS how the
    ↪  question asked in the evaluation can be answered based on the
    ↪ findings. Only include the key information.
- Use the JSON structure from the examples!

# Proceed with the Input:
**Input:**
```

```
924  INDICATION:  <IND>
925  FINDINGS:
926  <FIND>
927
928  **Output JSON:**
929
```

Listing 6: Few-shot example for indication extraction.

```
930
931
932  **Input:**
933  INDICATION: \_\_\_\_F with new onset ascites  // eval for infection
934  FINDINGS:
935  [S01] There is no focal consolidation, pleural effusion or
936      ↪ pneumothorax.
937  [S02] No acute cardiopulmonary process.
938
939  **Output JSON:**
940  {
941      "patient\_info": "female",
942      "indication": "New onset ascites.",
943      "evaluation": "Evaluate for infection.",
944      "indication\_summary": "Female with new onset ascites; should be
945          ↪ evaluated for infection.",
946      "associated\_findings": ["S02"],
947      "answer\_for\_indication": "Evaluation for infection is negative:
948          ↪ There is no acute cardiopulmonary process."
949  }
950
```

### E.1.3  Building Scene Graphs

**Entity Mapping**  We apply semantic entity mapping to modifiers (used to fill the attributes `probability`, `certainty`, `positiveness`, and `modifiers`), regions (attribute `regions`), finding entities (attribute `obs_entities`), and changes (attribute `changes`).

For each of these we consider the associated tags extracted by the LLM during observation extraction and encode them into text embeddings using the BioLORD [29] model. We also encode all potential tags and their synonyms, defined for each type of tag in our reference definitions. Then we compute the cosine similarities of each tag with all reference tags of the same type. We pick the reference tag with the highest cosine similarity but threshold it at $0.5$. If no reference tag was identified with cosine similarity $\geq 0.5$, then we mark the tag as non-resolved. For finding entities, we follow a slightly more complicated matching approach. Instead of only considering the finding entity tags extracted by the LLM, we also consider pairs of these entities and extracted region tags as well as the extracted summary sentences and names for matching. We then try to match each of those with the reference finding tags and pick the ones with the highest cosine similarities.

The matched reference finding tags are stored in the `obs_entities` attribute (non-resolved ones are kept in `non_resolved_obs_entities`), matched reference regions are stored in the `regions` attribute, where we also store the distance as identified by the LLM (non-resolved regions are kept in `non_resolved_regions`). The matched changes are store in the `changes` attribute (non-resolved changes are discarded). For all modifiers, we use the modifier type defined for the matched reference tag. We matched all modifiers against all types of modifiers, which means that the modifier type identified by the LLM can be overwritten during matching. Finally, we extract the probability from the modifiers (this is a special modifier type), store it in the `probability` attribute and infer the `certainty` and `positiveness` attributes from it (using the reference definitions). The remaining modifiers are stored in the `modifiers` attribute (non-matched ones are discarded).

We additionally try to identify the laterality of the observations. Here, we do not use semantic entity mapping but rely on keywords instead. We consider the raw finding entities, regions, as well as the summary sentences, and search for any laterality-related mentions such as *left*, *right*, *bilateral*, and related terms. From this we infer the laterality and store it into the `laterality` attribute.

**Reference Data and Standardization** Using the reference definitions, we infer all `obs_entities_parents`, `obs_categories`, `obs_subcategories`, and `default_regions` from the matched `obs_entities`.

Next, we inspect the `summary_sentence` and `name` attributes (extracted by the LLM) for underscores or mentions of changes. We track such issues (which are used for quality assessment) but do not apply any cleanup. Similarly, we check the `change_sentence` for underscores and assert that it contains mentions of changes.

We further inspect the structure of observations and their children. If an observation mentions multiple different findings and has one child for each of these findings, then we lift these children to the top-level and discard the parent. Similarly, we merge multiple duplicate observations into one.

Finally, we try to resolve missing regions or improve their precision. If no regions could be extracted, we rely on the `default_regions` derived from the `obs_entities` instead, but consider the identified laterality. We also check whether these `default_regions` are more precise than extracted ones. Then we check whether any identified region contradicts the identified laterality and remove them. We then either split or merge bilateral versions of the same region.

**Graph Construction** Based on the matched regions, we associate bounding boxes with the observations if available. Additionally, we build a tree of all mentioned regions and fill missing intermediate regions based on the reference data. This allows us to build a graph of region nodes relevant to the study.

We construct `region_region_relations` based on the reference data alone. `located_at_relations` are constructed based on the `regions` attribute of observations (direct specified). Additionally, we infer `located_at_relations` relations for sub regions (`sub_region`) and bilateral versions of regions (`bilateral`). `obs_relations` are constructed based on the parent-child structure of observations and their child type, as predicted by the LLM. `obs_sent_relations` are constructed based on the sentences each observation was derived from.

Finally, we attach the indication information extracted from the report. Therefore, we build an additional observation node based on the LLM-extracted `answer_for_indication` and the LLM-extracted associated sentences, from which we can infer the associated observations and can infer all relevant tags.

### E.2 Question-Answer Generation

### E.2.1 Template Engine

To construct QA-pairs, we develop a template-engine that considers the information in the scene graphs to construct the answers. The template engine generates a QA-pair by running the following steps:

1. Filter observations and studies based on the template configuration, e.g. only keeping observations of specific sub-categories.

2. Run a QA-strategy (indication, study abnormality, region abnormality, or finding) on the remaining scene graph. The strategy provides multiple named subsets of observations, variables to fill the template, as well as an overall state consisting of multiple tags (e.g. is the study positive, are there any devices, . . . ).

3. Construct the template-based main answer by selecting and filling the answer-template based on the state returned by the QA-strategy and the returned variables. Tags and bounding boxes can be inferred from defined observation subsets. (Not all templates provide such main answers)

4. Pick observation subsets identified by the QA-strategy and convert the observations into answer parts. The template configuration defines which subsets are picked and how they are ordered. Additionally, template-based prefix- or fallback-answers can be defined for each subset. Some subsets can also be excluded based on the QA-strategy state. These answers can be main-answers, details, or related information as defined in the configuration.

Additionally, the template engine supports variables, i.e. each template can be used to generate multiple QA-pairs. Variables can either be defined as lists (configured in the template) or can be

provided by the QA-strategy (which might infer variables from the current scene graph, e.g. all mentioned regions). The question may then also contain such template variables.

### E.2.2 Strategies and Templates

**Indication** In this strategy, we use the extracted indication (if available) as the question. More precisely, we use the `indication_summary` attribute from the indication node as the question text. The main-answer is constructed from the indication observation (i.e. the answer to the indication based on the finding sentences), while detail answer parts are constructed based on all associated finding observations. We include this question, if an indication observation is present in the scene graph.

**Study abnormality** In this strategy, we generate questions about abnormalities. This includes descriptions of the full study or specific categories of observations (e.g. devices), description of only abnormal findings, and yes/no questions of whether there are positive findings (overall or of specific categories) present in the study. We use the templates defined in Tab. 10.

The strategy identifies five types of observations: (i) finding (positive), (ii) finding (negative), (iii) device (positive), (iv) device (negative), (v) acquisition. Based on the specific template, these are selected as main-answers, details, or related information. Additionally, a template answer can be included, which is selected based on whether the study is abnormal or not. Some templates use different subcategories as variables, i.e. one question is generated for each of the defined subcategories, where observations are filtered based on this subcategory.

Table 10: Study abnormality templates.

| Template (ID) | Question Example | Variables | Main answer | Details | Related Inf. |
|---|---|---|---|---|---|
| describe_all
B01_describe_all | Describe the given study. | – | finding (positive)
device (positive)
device (negative)
finding (negative)
acquisition | – | – |
| describe_abnormal
B02_describe_abnormal | Describe all abnormal findings in the given study. | – | finding (positive) | – | device (positive) |
| is_abnormal
B03_is_abnormal | Are there any abnormal findings? | – | template
finding (positive) | finding (negative) | device (positive)
finding (negative) |
| is_normal
B04_is_normal | Is the study normal? | – | template | finding (positive)
finding (negative) | finding (negative)
device (positive) |
| describe_subcat
B08_describe_subcat | Evaluate the cardiac structures. | subcategory | finding (positive)
finding (negative) | – | – |
| describe_abnormal_subcat
B09_describe_abnormal_subcat | Describe any pulmonary diseases and disorders suggested by the study. | subcategory | finding (positive) | – | – |
| is_abnormal_subcat
B10_is_abnormal_subcat | Are there any fractures or bone diseases apparent from the study? | subcategory | template
finding (positive) | finding (negative) | finding (negative) |
| is_normal_subcat
B11_is_normal_subcat | Are the mediastinal and hilar contours normal? | subcategory | template
finding (positive) | finding (negative) | finding (negative) |
| describe_device
B12_describe_device | Check the presence and position of devices, tubes, lines, and other foreign objects. | subcategory | device (positive)
device (negative) | – | – |
| has_devices
B13_has_devices | Are there any signs of prior surgical procedures? | subcategory | template
device (positive) | device (negative) | device (negative) |
| describe_acquisition
B14_describe_acquisition | Assess the image quality and describe aspects related to image acquisition. | – | acquisition | – | – |
| describe_imaging_artifacts
B15_describe_imaging_artifacts | Describe any apparent imaging artifacts and imaging-related shadows. | – | acquisition | – | – |
| has_imaging_artifacts
B16_has_imaging_artifacts | Are there any imaging artifacts or imaging-related shadows? | – | template
acquisition | – | – |

**Region abnormality**   In this strategy, we generate question about anatomical regions. This includes describing regions, answering yes/no questions about the abnormality of regions, or describing specific aspects of regions (e.g. devices). We use the templates defined in Tab. 11.

For a given region, the strategy first identifies observations associated with that region and classifies them into the five types defined in the study abnormality strategy. Additionally, it identifies observations in related regions. This includes positive findings in parent regions or the opposite laterality. Additionally, a template answer can be included, which is selected based on whether the region is abnormal or not.

Before generating QA-pairs, the strategy first identifies a set of regions. For each of these regions an individual QA-pair is generated. The set of regions is computed as follows: We always include a set of pre-defined default regions (the lungs, the heart, . . . ) and include all regions explicitly mentioned in observations, as well as their parent regions. Additionally, we randomly sample regions. Their sampling probabilities are computed based on how often they are associated with positive vs. negative findings, i.e. the more often a region is associated with positive findings and the less often it is associated with negative findings, the more often we sample it as a question. This assures that we generate additional negative questions for regions that are only/mostly mentioned with positive findings.

Table 11: Region templates.

| Template (ID) | Question Example | Variables | Main answer | Details | Related Inf. |
|---|---|---|---|---|---|
| describe_region
C01_describe_region | Describe the left lung. | region | finding (positive)
device (positive)
finding (negative)
device (negative) | – | related regions |
| describe_abnormal_region
C02_describe_abnormal_region | Describe all abnormal findings in the lung bases. | region | finding (positive) | – | device (positive)
related regions |
| is_abnormal_region
C03_is_abnormal_region | Are there any abnormal findings in the mediastinum? | region | template
finding (positive) | device (positive)
finding (negative) | related regions |
| is_normal_region
C04_is_normal_region | Is the heart normal? | region | template
finding (positive) | region (positive) | finding (negative)
related regions |
| describe_region_device
C07_describe_region_device | Check the right chest for implants. | region
subcategory | device (positive)
device (negative) | – | related regions |
| has_region_device
C08_has_region_device | Are there any tubes, lines, or ports in or near the left lung? | region
subcategory | template
device (positive) | device (negative) | device (negative)
related regions |

**Finding**   In this strategy, we generate question about specific findings (radiological findings, diseaes, devices, . . . ). This includes descriptions of findings, yes/no questions about the presence of findings, location of findings, and severity of findings. We use the templates defined in Tab. 12.

For a given finding/device entity, the strategy first identifies observations associated with it and classifies them into the five types defined in the study abnormality strategy. Additionally, it identifies observations that contain related finding/device entities. This includes parent findings (i.e. findings that are parents of the current one), same subcat findings (i.e. findings having the same sub-category), correlated findings (based on statistics computed over the whole scene graph dataset), indications of the current finding, and findings that are indicative of the current finding. The observation subset can be selected based on the template configuration. Additionally, a template answer can be included, which is selected based on whether the finding is present or not and based on severity levels. This template may also be filled with information about the localization of the finding.

Before generating QA-pairs, the strategy first identifies a set of finding/device entities. For each of these entities an individual QA-pair is generated. The set of entities is computed as follows: We always include a set of pre-defined default entities and include all entities explicitly mentioned in observations, as well as their parent entities. Additionally, we randomly sample entities. Their sampling probabilities are computed based on how often they are mentioned positively vs. negatively (over all scene graphs), i.e. the more often a finding is mentioned positively and the less often it is mentioned negatively, the more often we sample it as a question. This assures that we generate additional negative questions for findings that are only/mostly mentioned positively.

Table 12: Finding templates.

| Template (ID) | Question Example | Variables | Main answer | Details | Related Inf. |
|---|---|---|---|---|---|
| describe_finding
D01_describe_finding | Describe the pleural effusion. | finding | finding (positive)
finding (negative) | – | parent findings
indications
indicative of
same subcat
correlated |
| has_finding
D02_has_finding | Is there any indication of pneumonia? | finding | template | finding (positive)
finding (negative) | parent findings
indications
same subcat
correlated |
| where_is_finding
D03_where_is_finding | Where is the lung nodule located? | finding | template | finding (positive)
finding (negative) | parent findings
indications |
| how_severe_is_finding
D04_how_severe_is_finding | How severe is the cardiomegaly? | finding | template | finding (positive)
finding (negative) | parent findings
indications |
| describe_device
D05_describe_device | Describe the endotracheal tube. | device | device (positive)
device (negative) | – | parent findings
same subcat |
| has_device
D06_has_device | Is a pacemaker visible in the study? | device | template | device (positive)
device (negative) | same subcat |
| where_is_device
D07_where_is_device | Where are the surgical clips located? | device | template | device (positive)
device (negative) | same subcat |

## E.3 Quality Assessment

**Scene Graph Quality** The scene graph quality aspects are computed by simply inspecting the observations nodes and checking which fields are set or empty. Additionally, we track issues during the graph construction procedure and derive quality aspects from them.

**QA Quality** We automatically assess the quality of the textual content of QA-pairs using Llama 3.1 8B [26] as a judge for the five criteria presented in Tab. 9.

For rating *entailment* (Listing 7), we condition the model on the report, the question, as well as the answer parts and we rate each answer part individually.

Listing 7: LLM prompt used for entailment evaluation of generated answers.

```
You will be given a Report (medical report of a chest X-ray study), a
    ↪ Question (about the study), and an Answer (to the question)
    ↪ consiting of several (numbered) sentences.

Your task is to assess/rate whether each of the answer sentences is
    ↪ true or not, given a the reference report about the chest X-ray
    ↪ . This task is known as entailment verification.
Assess the quality of each answer sentence independently and use one
    ↪ of the rating options provided below to assess how well the
    ↪ facts in each sentences align with the report.

# Guidelines:
- Rate each sentence in the Answer individually; do NOT use any prior
    ↪ answer sentences as context or source
- Provide the rating for each answer sentences in its own line
    ↪ starting with the sentence number followed by your rating
- For each sentence, use ONE of the rating options provided below, do
    ↪ NOT use any other options
- An example format will be provided
- DO NOT REPEAT the question or answer sentences in your response!

## Rating Options -- ONLY USE ONE OF THE FOLLOWING OPTIONS
- ALIGNED_MENTIONED: Answer aligns with the report (is factually
    ↪ correct) and all facts are explicitly stated in the report.
    Example: The same finding is described in the answer and the
        ↪ report
```

```
- ALIGNED_INFERABLE: Answer aligns with the report (is factually
    ↪ correct) but some facts are NOT explicitly stated in the report
    ↪ , can however be derived from what is written there.
    Example: The answer provides a more general description of what is
        ↪  written in the report.
- ALIGNED_NEGATIVE_NOT_MENTIONED: Answer does NOT contradict the
    ↪ report (may factually correct) but some facts (negative
    ↪ findings) cannot be derived from the report, are however likely
    ↪  correct because they are negative findings and nothing
    ↪ contradictory is mentioned in the report.
    Example: The answer mentions that a finding is not present but
        ↪ this is not explicitly mentioned and does not contradict
        ↪ anything in the report.
- ALIGNED_GENERAL_STATEMENT: Answer is a more general statement or
    ↪ summary that is not explicitly mentioned but aligns roughly
    ↪ with the overall report.
    Example: Summaries of whether the study is positive or negative.
- NON_ALIGNED_NON_INFERABLE: Answer does NOT contradict the report but
    ↪  the correctneaa of some facts cannot be validated using the
    ↪ report.
    Example: The answer mentions that a finding is present but this is
        ↪  never mentioned in the report and cannot be concluded from
        ↪  it.
- NON_ALIGNED_MISLEADING: Answer does NOT directly contradict the
    ↪ report but the description is highly misleading considering the
    ↪  report.
    Example: The answer mentions that a finding is not present, which
        ↪ is never mentioned in the report but could likely be
        ↪ present considering the report.
- NON_ALIGNED_CONTRADICTING: Answer contradict with the report.
    Example: The answer describes that a finding is not present, which
        ↪  is however mentioned as present in the report or vice
        ↪ versa.

# Example Format:
Report:
--- START OF REPORT ---
...
--- END OF REPORT ---

Question: ...

Answer (2 sentences to rate):
[01] First answer sentence.
[02] Second answer (last sentence in this example).

Rating (provide 2 ratings):
[01] ALIGNED_MENTIONED
[02] NON_ALIGNED_NON_INFERABLE

# Proceed with the following Report, Question, and Answer sentences:
Report:
--- START OF REPORT ---
 <REPORT>
--- END OF REPORT ---

Question:  <QUEST>

Answer ( <NUMANS>  sentences to rate):
 <ANSWERS>

Rating (provide  <NUMANS>  ratings):
```

For rating *relevance* (Listing 8), we condition the model on the question as well as the answer parts (but not on the report) and we rate each answer part individually.

Listing 8: LLM prompt used for relevance evaluation of generated answers.

```
You will be given a question (about a chest X-ray study), and an
    ↪ answer (to the question) consiting of several (numbered)
    ↪ sentences.

Your task is to assess/rate whether each of the answer sentences
    ↪ relevant to answer the question or is redundant.
Assess the quality of each answer sentence and use one of the rating
    ↪ options provided below.

# Guidelines:
- Rate each sentence in the answer individually; but check for
    ↪ redundancy with previous sentences.
- Provide the rating for each answer sentences with the sentence
    ↪ number followed by your rating.
- For each sentence, use ONE of the rating options provided below, do
    ↪ NOT use any other options.
- An example format will be provided.
- DO NOT REPEAT the question or answer sentences in your response!

## Rating Options -- ONLY USE ONE OF THE FOLLOWING OPTIONS
- RELEVANT_MAIN_ANSWER: Fullfill ALL of the following
    a) Are relevant to the question
    b) Are needed to answer the question or provide details
    c) Are not redundant to previous RELEVANT_MAIN_ANSWER sentences
- RELATED_INFO:  Fullfill ALL of the following
    a) Are NOT needed to answer the question
    b) Provide additional context related to the question or other
        ↪ answer sentences
    c) Are not redundant to any previous sentences
- REDUNDANT_INFO(...): Fullfill ALL of the following
    a) Would fullfill criteria a-b) for RELEVANT_MAIN_ANSWER, or
        ↪ RELATED_INFO
    b) Contains exactly the same information that was already provided
        ↪  in a previous sentence of the same type (ONLY consider
        ↪ previous sentences here!)
    c) Does not provide any additional details or related information
    d) Could be removed without changing the content of the answer
    Note: replace ... with the sentences IDs OF PREVIOUS SENTENCE with
        ↪  which the current sentence is redundant
- IRRELEVANT_INFO: Fullfill ALL of the following
    a) Does not classify as any of the above
    b) No information in the sentence is relevant or related to the
        ↪ question

# Example Format:
Question: ...

Answer (4 sentences to rate):
[01] First answer sentence.
[02] Second answer.
[03] Third answer, containint no additional information, everything
    ↪ was already mentioned in 01 and 02.
[04] Fourth sentence.

Rating (provide 4 ratings):
[01] RELEVANT_MAIN_ANSWER
[02] IRRELEVANT_INFO
[03] REDUNDANT_INFO(01,02)
[04] RELATED_INFO
```

```
1251
1252  # Proceed with the following Question and Answer sentences:
1253  Question:  <QUEST>
1254
1255  Answer ( <NUMANS>  sentences to rate):
1256   <ANSWERS>
1257
1258  Rating (provide  <NUMANS>  ratings):
1259
```

For rating *completeness* (Listing 9), we condition the model on the report, the question, as well as the full answer and we rate the full answer as a whole.

Listing 9: LLM prompt used for completeness evaluation of generated answers.

```
1262
1263  You will be given a Report (medical report of a chest X-ray study), a
1264      ↪ Question (about the study), and an Answer (to the question) .
1265
1266  Your task is to assess/rate whether the provided Answer contains all
1267      ↪ the necessary information to answer the Question, considering
1268      ↪ the Report as the source of truth.
1269
1270  # Guidelines:
1271  - Do not assess whether the answer is correct but whether it is
1272      ↪ contains all relevant information from the Report to answer the
1273      ↪  Question.
1274  - Use ONE of the rating options provided below, do NOT use any other
1275      ↪ options.
1276  - Answer with a short explanation (a few words) followed by "->" and
1277      ↪ the rating option.
1278  - An example format will be provided.
1279  - DO NOT REPEAT the report, question, or answer sentences in your
1280      ↪ response!
1281
1282  ## Rating Options -- ONLY USE ONE OF THE FOLLOWING OPTIONS
1283  - FULLY_COMPLETE: All facts from the report that are relevant to the
1284      ↪ question are included and the question is answered.
1285  - DETAILS_MISSING: The main facts from the report that are relevant to
1286      ↪  the question are included BUT some details are missing.
1287  - NOT_ANSWERED: While facts from the report may be contained, the
1288      ↪ answer does not relate to the question at all.
1289  - INCOMPLETE_NON_MISLEADING: Main facts are missing, but these should
1290      ↪ not lead to a misrepresentation of the facts (e.g. only some
1291      ↪ negative findings are not mentioned).
1292  - INCOMPLETE_MISLEADING: Important facts are missing, such that the
1293      ↪ answer may mislead the reader.
1294
1295  # Example Format:
1296  Report:
1297  --- START OF REPORT ---
1298  ...
1299  --- END OF REPORT ---
1300
1301  Question: ...
1302
1303  Answer (to rate):
1304  ...
1305
1306  Rating (your task):
1307  severity is missing -> DETAILS_MISSING
1308
1309
1310  # Proceed with the following Report, Question, and Answer:
1311  Report:
1312  --- START OF REPORT ---
1313   <REPORT>
```

```
1314  --- END OF REPORT ---
1315
1316  Question:  <QUEST>
1317
1318  Answer (to rate):
1319   <ANSWERS>
1320
1321  Rating (your task):
1322
```

For rating *question clarity* (Listing 10), we condition the model on the question only and rate it.

Listing 10: LLM prompt used for question clarity evaluation of generated questions.

```
1324
1325  You will be given a medical Question about a radiological chest X-ray
1326      ↪ study (which is not provided).
1327  Your task is to assess/rate the clarity of the Question, i.e. whether
1328      ↪ its wording is clear and unambiguous, and whether it is easy to
1329      ↪  understand and answer.
1330
1331  # Guidelines:
1332  - Use ONE of the rating options provided below, do NOT use any other
1333      ↪ options
1334  - Answer with a short explanation (a few words) followed by "->" and
1335      ↪ the rating option
1336  - An example format will be provided
1337  - DO NOT REPEAT any part of the question or answer sentences in your
1338      ↪ response!
1339
1340  ## Rating Options -- ONLY USE ONE OF THE FOLLOWING OPTIONS
1341  - OPTIMAL: The question is mostly clear, unambiguous, and can be
1342      ↪ answered. It is well-structured and concise without grammatical
1343      ↪  errors.
1344  - UNUSUAL_SENTENCE_STRUCTURE: The question is mostly clear,
1345      ↪ unambiguous, and can be answered. However, the sentence
1346      ↪ structure is unusual or complex. There are no grammatical
1347      ↪ errors.
1348  - GRAMMATICAL_ERRORS: The question is mostly clear, unambiguous, and
1349      ↪ can be answered. However, there are grammatical errors that may
1350      ↪  affect the clarity. The sentence may or may not be well-
1351      ↪ structured.
1352  - UNRELATED_TO_CHEST_XRAY: The question is mostly clear and
1353      ↪ unambiguous. However, it does not make sense to ask this
1354      ↪ question about a chest X-ray study, because it does not relate
1355      ↪ to the content that can be observed in a chest X-ray. There may
1356      ↪  or may not be grammatical errors or unusual sentence structure
1357      ↪ .
1358  - UNCLEAR_QUESTION: The question may be misunderstood, is ambiguous,
1359      ↪ or otherwise unclear. Any answer could be misleading or
1360      ↪ incorrect, even with proper medical knowledge and context.
1361      ↪ There may or may not be grammatical errors or unusual sentence
1362      ↪ structure.
1363
1364  Note that simply stating the indication/history motivating the study
1365      ↪ is considered a valid question (and should not be rated as
1366      ↪ UNCLEAR_QUESTION solely for not being an explicit question)!
1367
1368  # Example Format:
1369  Question (to rate): ...
1370
1371  Rating (your task):
1372  The question is unrelated to chest X-rays -> UNRELATED_TO_CHEST_XRAY
1373
1374
1375  # Proceed with the following Question:
1376  Question (to rate):  <QUEST>
```

```
1377
1378    Rating (your task):
1379
```

For rating *answer clarity* (Listing 11), we condition the answer parts only (but not on the report or question) and we rate each answer part individually.

Listing 11: LLM prompt used for answer clarity evaluation of generated answers.

```
1382
1383    You will be given a medical Answer to an unknown question about a
1384       ↪ radiological chest X-ray study (which is not provided).
1385    Your task is to assess/rate the clarity of each sentence of the Answer
1386       ↪ , i.e. whether its wording is clear and unambiguous, and
1387       ↪ whether it is easy to understand.
1388
1389    # Guidelines:
1390    - Rate each sentence in the Answer individually; do NOT use any prior
1391       ↪ answer sentences as context or source
1392    - Provide the rating for each answer sentences in its own line
1393       ↪ starting with the sentence number followed by your rating
1394    - For each sentence, use ONE of the rating options provided below, do
1395       ↪ NOT use any other options
1396    - An example format will be provided
1397    - DO NOT REPEAT the question or answer sentences in your response!
1398
1399    ## Rating Options -- ONLY USE ONE OF THE FOLLOWING OPTIONS
1400    - OPTIMAL: The answer sentence is mostly clear and unambiguous. It is
1401       ↪ well-structured and concise without grammatical errors.
1402    - UNUSUAL_SENTENCE_STRUCTURE: The answer sentence is mostly clear and
1403       ↪ unambiguous. However, the sentence structure is unusual or
1404       ↪ complex. There are no grammatical errors.
1405    - GRAMMATICAL_ERRORS: The answer sentence is mostly clear and
1406       ↪ unambiguous. However, there are (severe) grammatical errors
1407       ↪ that affect the clarity. The sentence may or may not be well-
1408       ↪ structured.
1409    - UNCLEAR_ANSWER: The answer sentence may be misunderstood, is
1410       ↪ ambiguous, or otherwise unclear. There or may not be
1411       ↪ grammatical errors or unusual sentence structure.
1412    - NOT_UNDERSTANDABLE: The answer sentence cannot be understood at all.
1413       ↪  It is completely unclear, nonsensical, gibberish, or
1414       ↪ contradictory in itself. There may or may not be grammatical
1415       ↪ errors or unusual sentence structure.
1416
1417    # Example Format:
1418    Answer (4 sentences to rate):
1419    [01] This first sentence.
1420    [02] This is the second answer sentence.
1421    [03] Some text where it is unclear what is meant.
1422    [04] This is the last answer sentence.
1423
1424    Rating (provide 4 ratings):
1425    [01] GRAMMATICAL_ERRORS
1426    [02] OPTIMAL
1427    [03] UNCLEAR_ANSWER
1428    [04] OPTIMAL
1429
1430
1431    # Proceed with the following Answer sentences:
1432    Answer ( <NUMANS>  sentences to rate):
1433     <ANSWERS>
1434
1435    Rating (provide  <NUMANS>  ratings):
1436
```

### E.4 Resources for Dataset Construction and Evaluation

#### E.4.1 Source Datasets

**MIMIC-CXR [14], [15], [21]**  We use the MIMIC-CXR dataset version 2.1.0 (`https://physionet.org/content/mimic-cxr/2.1.0/` as the source of radiology reports from which we extract the scene graphs. It contains 227 835 radiographic (chest X-ray) studies performed at the Beth Israel Deaconess Medical Center in Boston, MA, USA. It is licensed under the PhysioNet Credentialed Health Data License 1.5.0.

**MIMIC-CXR-JPG [15], [24], [25]**  We use the MIMIC-CXR-JPG dataset version 2.1.0 (`https://physionet.org/content/mimic-cxr-jpg/2.1.0/`) as the source of images for localization (the CXAS segmntation model is applied on these images). Additionally, we use the provided radiologist annotations (`mimic-cxr-2.1.0-test-set-labeled.csv`) as targets to evaluate the quality of extracted finding tags (Tabs. 2a and 4). The dataset is derived from MIMIC-CXR and is licensed under the PhysioNet Credentialed Health Data License 1.5.0.

**Chest ImaGenome [15], [17], [19]**  We use the Chest ImaGenome Dataset version 1.0.0 (`https://physionet.org/content/chest-imagenome/1.0.0/`) as a source of anatomical region bounding boxes for localization. Additionally, we use their provided scene graphs as a baseline for the evaluations of our scene graphs (Tabs. 2 and 4 to 7). It contains scene graphs for 242 072 frontal images from MIMIC-CXR that have been created using rule-based natural language processing and CXR atlas-based bounding box detection. The dataset is derived from MIMIC-CXR and is licensed under the PhysioNet Credentialed Health Data License 1.5.0.

**CXR-LT 2024 [15], [30], [34]**  We use the CXR-LT 2024 dataset version 2.0.0 (`https://physionet.org/content/cxr-lt-iccv-workshop-cvamd/2.0.0/`) as targets to evaluate the quality of extracted finding tags (Tabs. 2a and 5). More precisely, we use the gold standard dataset provided for Task 2 in the CXR-LT 2024 challenge tasks (406 reports, 26 classes). The dataset is derived from a small subset of MIMIC-CXR and was hand-labeled by radiologists. It is licensed under the PhysioNet Credentialed Health Data License 1.5.0.

**MS-CXR [15], [31], [35]**  We use the MS-CXR dataset version 1.1.0 (`https://physionet.org/content/ms-cxr/1.1.0/`) as targets to evaluate the quality of extracted finding boxes (Tabs. 2b and 6). The dataset contains 1162 image-sentence pairs of bounding boxes and corresponding phrases (and their finding classes) for 8 different findings. It is derived from a small subset of MIMIC-CXR and was hand-labeled by radiologists. It is licensed under the PhysioNet Credentialed Health Data License 1.5.0.

**REFLACX [15], [32], [36]**  We use the REFLACX dataset version 1.0.0 (`https://physionet.org/content/reflacx-xray-localization/1.0.0/`) as targets to evaluate the quality of extracted finding boxes (Tabs. 2b and 7). The dataset provides eye-tracking data collected for 3032 frontal chest x-rays from the MIMIC-CXR dataset. Additionally, it provides hand-labeled ellipses localizing for several anomalies present in the images. We only use the ellipses but do not use the eye-tracking data. It is licensed under the PhysioNet Credentialed Health Data License 1.5.0.

#### E.4.2 Models

**Llama 3.1 70B [26]**  We use the AWQ-INT4 [40] quantized version of Llama 3.1 70B Instruct provided by the Huggingface hub at `https://huggingface.co/hugging-quants/Meta-Llama-3.1-70B-Instruct-AWQ-INT4`. The model is derived from the `https://huggingface.co/meta-llama/Llama-3.1-70B-Instruct` and is licensed under the LLAMA 3.1 COMMUNITY LICENSE AGREEMENT. We limit the maximum number of tokens to 6144.

**Llama 3.1 8B [26]**  We use the AWQ-INT4 [40] quantized version of Llama 3.1 70B provided by the Huggingface hub at `https://huggingface.co/hugging-quants/Meta-Llama-3.1-8B-Instruct-AWQ-INT4`. The model is derived from the `https://huggingface.co/meta-llama/Llama-3.1-8B-Instruct` and is licensed under the LLAMA 3.1 COMMUNITY LICENSE AGREEMENT. We limit the maximum number of tokens to 8192.

**CXAS [22], [23]** We use the model provided by the CXAS Python library `https://pypi.org/project/cxas/`. See also `https://github.com/ConstantinSeibold/ChestXRayAnatomySegmentation`. It is licensed under the Attribution-NonCommercial-ShareAlike 4.0 International license. We run segmentation of all anatomical structures on half the original image resolution (half original image width and height).

**BioLORD [29]** We use the `BioLORD-2023-C` variant provided by the Huggingface model hub at `https://huggingface.co/FremyCompany/BioLORD-2023-C` and licensed under the MIT license. To apply the model, we use the Sentence Transformers library (`https://github.com/UKPLab/sentence-transformers`), which is licensed under the Apache-2.0 license.

**Model Inference Details** For all LLM-based information extraction steps, we rely on the vLLM library [41] (`https://github.com/vllm-project/vllm`, Apache-2.0 license) for inference. We run all models with $temperature = 0.0$. All json-outputs are parsed using the Pydantic libary (`https://docs.pydantic.dev`).

### E.4.3 Computational Costs

Each dataset construction step can run on an individual Nvidia A100 GPU, but we use multiple GPUs in parallel, with each GPU responsible for a different subset of the dataset. Semantic segmentation of all 158 anatomical structures using the CXAS models takes about 6 seconds per image, leading to a total of about 628 GPU hours. Sentence extraction takes about 1 second per study (report), leading to a total of about 65 GPU hours (for 227 835 studies). Observation extraction takes about 1.7 seconds per study, leading to a total of about 108 GPU hours. Indication extraction takes about 0.3 seconds per study, leading to a total of about 19 GPU hours. Scene graph construction (including entity matching) takes about 0.6 seconds per study, leading to a tool of about 38 hours. Question-answer generation does not require a GPU but takes about 9 seconds per study (including all question templates and strategies), leading to a total of about 24 days. However, multiple processes can be run in parallel on a single machine, leading to an effective time of only about a day for all 42M QA-pairs. Quality assessment of QA texts again requires a GPU and consists of 5 individual steps that can be run in parallel. Overall the assessment takes about 6 GPU days for all 42M QA-pairs.

### E.5 Dataset Release

We release the dataset as a credentialed dataset on the Physionet [15] platform (`https://physionet.org/`) under the PhysioNet Credentialed Health Data License 1.5.0 (`https://physionet.org/about/licenses/physionet-credentialed-health-data-license-150/`). This makes the dataset openly accessible to all researches credentialed by PhysioNet, which requires a short online training. This type of hosting is required because we derived our dataset from the MIMIC-CXR [14] dataset. Additionally, this is also a responsible safeguard to protect the data that is (indirectly) derived from patient health data. While enabling researchers access to the dataset, it limits the access for other purposes. Additionally, it requires researchers to complete a privacy and ethics course. Code to construct the dataset and to train on it is made openly available.

**Societal Impact** As a large vision-language dataset for medical imaging, this dataset has significant potential for societal impact. However, its use as a training source for models employed in clinical or medical applications also poses a substantial risk of misdiagnosis, highlighting the need for caution. Therefore, we strongly advise against relying solely on this dataset for fine-tuning or evaluating such models. On the other hand, this dataset can facilitate the development of large and interactive VQA models, which can provide supplemental information for patients, serve as a training tool for healthcare professionals, or optimize clinical workflows. The provided annotations, including bounding boxes and tags, further enhance its utility by providing a level of transparency and explainability in model predictions, allowing for more informed interpretation and analysis. By sparking research in this direction, this dataset can contribute to the advancement of the field and ultimately lead to positive long-term societal impacts. Nevertheless, it is essential to approach this dataset with caution, recognizing its limitations and potential risks if used improperly. As such, we consider this dataset a valuable research asset, but not yet suitable as a (sole) training source for real-world medical applications, emphasizing the need for careful evaluation and validation.

# F   Structured VQA Task

## F.1   Further Structured VQA Results

Table 13: Further results of our structured VQA task (Sec. 5). We show all the metrics from Tab. 3 with additional sub-metrics of our RadStrucVQA metric. Besides our default VQA model and the MAIRA-2 baseline, we also show alternative settings of our VQA model, namely training without bounding boxes and/or tags and predicting bounding boxes and tags before or after the text. Apart from these adaptions, the experimental setup was the same as in Sec. 5. We found that none of these adaptions has major influences on the results (apart from being capable of predicting boxes/tags), indicating that text, boxes, and tags in our dataset do not contradict each other. However, we observed minor improvements in text quality by adding bounding boxes

| | Model | Ours (ablations) | | | | Ours (default) | MAIRA-2 [18] |
|---|---|---|---|---|---|---|---|
| | Boxes | ✗ | ✓ | ✓ | ✓ | ✓ | ✓ |
| | Tags | ✗ | ✗ | ✗ | ✓ | ✓ | ✓ |
| | Text | only text | after text | before text | after text | before text | after text |
| RadFact* | Logical Prec.. | 0.75 | **0.76** | **0.76** | **0.76** | **0.76** | 0.25 |
| | Logical Rec.. | 0.74 | **0.75** | **0.75** | **0.75** | **0.75** | 0.64 |
| | Logical F1. | 0.74 | 0.74 | **0.75** | **0.75** | **0.75** | 0.27 |
| | Grounding Prec.. | – | **0.88** | 0.87 | **0.88** | 0.87 | 0.69 |
| | Grounding Rec.. | – | 0.88 | **0.90** | 0.88 | 0.89 | 0.12 |
| | Grounding F1. | – | **0.83** | **0.83** | **0.83** | **0.83** | 0.32 |
| | Spatial Prec.. | – | **0.68** | 0.67 | **0.68** | 0.67 | 0.12 |
| | Spatial Rec.. | – | 0.67 | **0.68** | **0.68** | **0.68** | 0.07 |
| | Spatial F1. | – | 0.63 | 0.63 | **0.64** | 0.63 | 0.06 |
| RadStrucVQA (Tags) | Finding Prec. | – | – | – | **0.68** | **0.68** | – |
| | Finding Rec. | – | – | – | **0.67** | 0.66 | – |
| | Finding F1 | – | – | – | **0.68** | 0.67 | – |
| | Finding-pos Prec. | – | – | – | 0.40 | **0.41** | – |
| | Finding-pos Rec. | – | – | – | **0.29** | 0.26 | – |
| | Finding-pos F1 | – | – | – | **0.39** | **0.39** | – |
| | Region Prec. | – | – | – | **0.67** | **0.67** | – |
| | Region Rec. | – | – | – | **0.66** | **0.66** | – |
| | Region F1 | – | – | – | 0.66 | **0.67** | – |
| | Region-pos Prec. | – | – | – | 0.29 | **0.34** | – |
| | Region-pos Rec. | – | – | – | **0.21** | **0.21** | – |
| | Region-pos F1 | – | – | – | 0.29 | **0.32** | – |
| | Main-category Prec. | – | – | – | **0.73** | **0.73** | – |
| | Main-category Rec. | – | – | – | **0.70** | **0.70** | – |
| | Main-category F1 | – | – | – | **0.72** | **0.72** | – |
| | Main-category-pos Prec. | – | – | – | 0.49 | **0.52** | – |
| | Main-category-pos Rec. | – | – | – | **0.36** | 0.34 | – |
| | Main-category-pos F1 | – | – | – | 0.47 | **0.49** | – |
| | Sub-category Prec. | – | – | – | **0.71** | **0.71** | – |
| | Sub-category Rec. | – | – | – | **0.67** | **0.67** | – |
| | Sub-category F1 | – | – | – | **0.69** | **0.69** | – |
| | Sub-category-pos Prec. | – | – | – | 0.47 | **0.50** | – |
| | Sub-category-pos Rec. | – | – | – | **0.34** | 0.32 | – |
| | Sub-category-pos F1 | – | – | – | 0.45 | **0.46** | – |
| | Bbox-pos-entity Prec. | – | – | – | 0.31 | **0.32** | – |
| | Bbox-pos-entity Rec. | – | – | – | **0.22** | 0.20 | – |
| | Bbox-pos-entity F1 | – | – | – | **0.26** | **0.26** | – |

*Our RadStrucVQA implementation.

## F.2 RadStrucVQA Metric

Following the RadFact [18] metric, we split the predictions into individual elements. In our case, we treat each answer part as its own element, ignoring the hierarchy level and order. For each QA-sample, this results in a set of prediction elements $\hat{\mathcal{Y}}$, where $|\hat{\mathcal{Y}}|$ is the number of answer parts in the predicted answer, and a set of target elements $\mathcal{Y}$, where $|\mathcal{Y}|$ is the number of answer parts in the target answer.

For each RadStrucVQ sub-metric (sub $\in \{\text{logical}, \text{grounding}, \text{finding}, \dots\}$), we compute a sample-level precision $p_{\text{sub}}$ and recall $r_{\text{sub}}$ score individually:

$$p_{\text{sub}}\left(\hat{\mathcal{Y}}, \mathcal{Y}\right) = s_{\text{sub}}\left(\hat{\mathcal{Y}}, \mathcal{Y}\right), \tag{1}$$

$$r_{\text{sub}}\left(\hat{\mathcal{Y}}, \mathcal{Y}\right) = s_{\text{sub}}\left(\mathcal{Y}, \hat{\mathcal{Y}}\right), \tag{2}$$

where $s_{\text{sub}}\left(\mathcal{H}, \mathcal{C}\right) \in [0, 1]$ is a sub-metric specific scoring function considering the hypothesis set $\mathcal{H}$ given the context set $\mathcal{C}$. For precision $\mathcal{H} = \hat{\mathcal{Y}}$ is the prediction set and $\mathcal{C} = \mathcal{Y}$ is the target set, while for recall $\mathcal{H} = \mathcal{Y}$ and $\mathcal{C} = \hat{\mathcal{Y}}$.

The score $s_{\text{sub}}$ is computed as the fraction of relevant hypothesis elements $h \in \mathcal{H}$ that are entailed, using a sub-metric specific entailment definition, given the context $\mathcal{C}$. More precisely:

$$s_{\text{sub}}\left(\mathcal{H}, \mathcal{C}\right) = \frac{\left|\left\{h \in \mathcal{H} \mid \text{entailed}_{\text{sub}}\left(h, \mathcal{C}[h]\right) \wedge \text{relevant}_{\text{sub}}(h)\right\}\right|}{\left|\left\{h \in \mathcal{H} \mid \text{relevant}_{\text{sub}}(h)\right\}\right|}, \tag{3}$$

where $\mathcal{C}[h]$ is the evidence from $\mathcal{C}$ for $h$ defined as

$$\mathcal{C}[h] = \left\{c \in \mathcal{C} \mid h \text{ is logically entailed with } \mathcal{C} \wedge c \text{ provides evidence for } h\right\}. \tag{4}$$

We compute $\mathcal{C}[h]$ by prompting an LLM to (i) identify entailment of $h$ given all context elements in $\mathcal{C}$, where $h$ can be `ENTAILED` or `NOT_ENTAILED` (neutral or contradicting); and (ii) provide the relevant evidence for entailment, i.e. the context units $c \in \mathcal{C}$ that support $h$. The LLM is given only the textual descriptions of each element (answer part), i.e. the entailment classification is purely logical and does not consider localization or any tags. Note that $\mathcal{C}[h] = \{\}$ if $h$ is not entailed.

Given the hypothesis $h$ and its evidence $\mathcal{C}[h]$, the sub-metric entailment is computed individually by

$$\text{entailed}_{\text{sub}}\left(h, \mathcal{C}[h]\right) \in \{\text{true}, \text{false}\}, \tag{5}$$

while the relevant subset of hypothesis elements is identified using the sub-metric specific

$$\text{relevant}_{\text{sub}}(h) \in \{\text{true}, \text{false}\}. \tag{6}$$

The definitions for each sub-metric can be found in Tab. 14.

Table 14: RadStrucVQA sub metric definitions. The logical, grounding, and spatial sub-metrics follow the same principles as the corresponding sub-metrics in RadFact [18].

| Sub-metric | $\mathrm{entailed_{sub}}(h, \mathcal{C}[h])$ | $\mathrm{relevant_{sub}}(h)$ |
|---|---|---|
| logical | $\mathcal{C}[h]$ is not empty
i.e. there is positive evidence for $h$ in $\mathcal{C}$
and $h$ does not contradict $\mathcal{C}$ | always $\mathtt{true}$ |
| grounding | $\mathrm{entailed_{logical}}(h, \mathcal{C}[h]) \wedge \mathrm{IoH}(h, \mathcal{C}[h]) \geq 0.5$
where IoH is the Intersection between boxes in $h$ and boxes in $\mathcal{C}[h]$
over the total box area in $h$,
with intersection/area computed based on box-masks (unions of boxes) | $h$ has bounding boxes $\wedge$
$\mathrm{entailed_{logical}}(h, \mathcal{C}[h])$ |
| spatial | $\mathrm{entailed_{grounding}}(h, \mathcal{C}[h])$ | $h$ has bounding boxes |
| finding | $\mathrm{entailed_{logical}}(h, \mathcal{C}[h]) \wedge$
each of the finding tags in $h$ is present in any of $\mathcal{C}[h]$,
only considering the subset of $\mathcal{C}[h]$ with the same positivity | $h$ has finding tags |
| finding-pos | $\mathrm{entailed_{finding}}(h, \mathcal{C}[h])$ | $\mathrm{relevant_{finding}}(h) \wedge$
$h$ is positive |
| region | $\mathrm{entailed_{logical}}(h, \mathcal{C}[h]) \wedge$
each of the region tags in $h$ is present in any of $\mathcal{C}[h]$,
only considering the subset of $\mathcal{C}[h]$ with the same positivity | $h$ has region tags |
| region-pos | $\mathrm{entailed_{region}}(h, \mathcal{C}[h])$ | $\mathrm{relevant_{region}}(h) \wedge$
$h$ is positive |
| main-category | $\mathrm{entailed_{logical}}(h, \mathcal{C}[h]) \wedge$
each of the finding main category tags in $h$ is present in any of $\mathcal{C}[h]$,
only considering the subset of $\mathcal{C}[h]$ with the same positivity | $h$ has finding main category tags |
| main-category-pos | $\mathrm{entailed_{main-category}}(h, \mathcal{C}[h])$ | $\mathrm{relevant_{main-category}}(h) \wedge$
$h$ is positive |
| sub-category | $\mathrm{entailed_{logical}}(h, \mathcal{C}[h]) \wedge$
each of the finding sub category tags in $h$ is present in any of $\mathcal{C}[h]$,
only considering the subset of $\mathcal{C}[h]$ with the same positivity | $h$ has finding sub category tags |
| sub-category-pos | $\mathrm{entailed_{sub-category}}(h, \mathcal{C}[h])$ | $\mathrm{relevant_{sub-category}}(h) \wedge$
$h$ is positive |
| bbox-pos-entity | $\mathrm{entailed_{finding}}(h, \mathcal{C}[h]) \wedge \mathrm{entailed_{grounding}}(h, \mathcal{C}[h])$ | $\mathrm{relevant_{finding-pos}}(h) \wedge$
$\mathrm{relevant_{spatial}}(h)$ |

**Implementation Details**    The final precision/recall scores are computed by averaging the sample-level scores. F1 scores can also be computed by first taking the per-sample harmonic mean of precision and recall before averaging the sample-level F1 scores. Invalid answers, samples with LLM parse errors during entailment computation, as well as samples without relevant hypotheses are ignored during averaging. We use the same entailment prompts and few-shot examples as in RadFact [18] but use the Llama 3.1 8B [26] model, allowing us to compute the metric locally.

## F.3  Experimental Setup

**Vision-Language Model Training**    Our vision-language model follows the Llava architecture [37], using Rad-DINO [38] (`microsoft/rad-dino`) for image encoding and the 3B Llama 3.2 language model (`https://huggingface.co/meta-llama/Llama-3.2-3B-Instruct`) connected via an MLP projection layer. We freeze the image encoder and all existing language model parameters but add new special tokens (with trainable embeddings) and apply LoRA [42] to the language model. Therefore, we only train the projection layer, the LoRA parameters, and the newly added token embeddings (keeping the existing token embeddings frozen). We train for one epoch on 1M samples from our CXR-QBA fine-tuning grade dataset (MIMIC-CXR's train split), where we use autoregressive training but only apply the loss to answer tokens. For image encoding and projection, we adopt the hyperparameters of MAIRA-2 [18]: We square-crop the images and resize them to $518 \times 518$, leading to $37 \times 37 = 1369$ image patches (i.e. image tokens), then we use the features of the last image encoder layer, and project the image tokens using 4 projection layers with GeLU activations. For LoRA, we use $r = 64$, $\alpha = 16$, and dropout 0.05. The maximum number of tokens for the language model is restricted to 2048. We use the AdamW optimizer with cosine annealing scheduling with 500 warmup steps, maximum learning rate $1e-3$, no weight decay, a batch size

of 4 with 16 accumulation steps, gradient norm clipping at 1.0, and `bf16` precision. The model is evaluated on the test split (following MIMIC-CXR) of our CXR-QBA fine-tuning grade set.

**Prompt and Special Tokens** Our question prompt follows the template shown in Listing 12, where `<boi>` (begin of image), `<eoi>` (end of image), and `<imgref1>` (first image reference) are newly added special tokens, `` tokens are replaced by image token features, and `{QUESTION}` is replaced by the specific question.

Listing 12: Question prompt.

```
Consider the following chest X-ray image: <boi><imgref1>...<eoi>
    ↪ {QUESTION}
```

The answers are formatted into sequences using XML-style structures and special tokens to represent tags and bounding boxes. An example is given in Listing 13.

Listing 13: Answer prompt.

```
<answer>
  <regions><bilateral><lungs></regions>
  <probability><certain><neg><probability>
  <categories>
    <ANATOMICAL_FINDING><DISEASE>
    <subcat>LUNG FIELD</subcat><subcat>PULMONARY DISEASES</subcat>
  </categories>
  <entities><entity>pneumothorax</entity></entities>
  <modifiers></modifiers>
  <box><imgref1><x51><y18><x90><y87><box>
  <box><imgref1><x09><y19><x52><y93></box>
  No, there is no indication of pneumothorax.
</answer>
```

We use special start and end tokens for answer parts (`<answer>` / `</answer>`), bounding boxes (`<box>` / `</box>`), and groups of tags (`<regions>` / `</regions>`, `<probability>` / `</probability>`, `<categories>` / `</categories>`, `<entities>` / `</entities>`, `<modifiers>` / `</modifiers>`). For some tags we use individual special tokens, namely for laterality (e.g. `<bilateral>`), regions (e.g. `<lungs>`), certainty (e.g. `<certain>`), positivity (e.g. `<neg>`), and main categories (e.g. `<ANATOMICAL_FINDING>`). For others we use start/end tokens and normal text, namely for sub-categories (`<subcat>` / `</subcat>`) and finding entities (`<entity>` / `</entity>`). Bounding boxes are listed after all other tags, where we use `<box>` / `</box>` tokens and refer back to the image using `<imgref1>`. Inside the box-tokens we use special relative coordinate tokens (following MAIRA-2 [18]) that represent the normalized $(x_1, y_1, x_2, y_2)$ coordinates of the bounding box, each quantized to 100 different tokens per dimension. We use different tokens for the $x$- and $y$-dimensions but share them for both corners (e.g. $x_1$ and $x_2$ share the same token set). The textual description is the last part of each answer part and consists of plain text without special tokens. If an answer consists of multiple answer parts, then each answer part uses an individual block as in Listing 13. All new token embeddings are initialized close to the existing token embeddings, where we try to initialize them based on keywords defined for each token. More precisely, given a set of keywords for a new token, we tokenize the keywords using the old vocabulary and compute the average embedding of all these tokens. This is then used as the initialization for the new token.

**MAIRA-2 Baseline** We use the MIARA-2 [18] checkpoint available at `https://huggingface.co/microsoft/maira-2`. We freeze the full model but modify the prompt. More precisely, we use their original prompt for grounded report generation but slightly modify it, asking the model to answer to the question (included in the modified prompt) instead of reporting all findings in the image. The rest of the prompt is kept unchanged. This model is then evaluated on the same test set as our vision-language model. It is capable of generating individual answer parts, each with bounding boxes, but does not generate bounding boxes for negative answers and cannot generate any tags.

**Computation Costs** We train on a single Nvidia A100 GPU (with 48GB of memory) for about 8 GPU days.

