# OpenReview forum: "A Structured, Tagged, and Localized Visual Question Answering Dataset with Full Sentence Answers and Scene Graphs for Chest X-ray Images"
_NeurIPS.cc/2025/Datasets_and_Benchmarks_Track — Submitted to NeurIPS 2025 Datasets and Benchmarks Track_

### Official Review · Reviewer_5Yi9 · 2025-06-24

**Rating:** 5
**Confidence:** 4

**Summary:**

This paper introduces CXR-QBA, a large-scale chest X-ray VQA dataset with 42 million QA-pairs featuring multi-part answers, bounding boxes, and structured tags, automatically generated from MIMIC-CXR using LLM-based information extraction from radiology reports. This dataset represents the largest and most sophisticated VQA resource for chest X-rays with impressive scale and annotation richness.

**Additional Feedback:**

### Rating
Quality: **3 (Good)**
The work is technically competent but has significant limitations in validation methodology, particularly the over-reliance on LLM-based assessment without human validation for a medical domain application.

Clarity: **3 (Good)**
While generally well-organized, the text contains multiple formatting issues and has unclear explanations of some concepts.

Significance: **2 (Fair)**
The dataset represents a substantial contribution to the medical VQA field in terms of scale and sophistication, though clinical applicability is limited by the lack of human validation.

Originality: **3 (Good)**
The work demonstrates a novel application of existing techniques at an unprecedented scale for medical VQA, though the core methodological approach builds heavily on established LVLM capabilities. The proposed task, Structured VQA, is interesting.

Generally, this is a good paper. But a solely automatically generated dataset limits the benefit of this dataset to the medical community. If the authors could provide a gold-standard dataset, I would further raise my score in Significant.
Note that I will also raise my scores in other criteria if the authors can address my concerns. However, improving other criteria may not raise my final Rating to 5, as an automatically generated dataset is very limited in the medical domain.

**Dataset Code Accessibility:**

Partly

**Dataset Code Comments:**

Generally the code and document are very detailed with many utilities for future researchers. But there are some issues. Below, I present some of the problems and my hotfixes.

**Missing reference data files**
```bash
cxr_qba_extract status # bug FileNotFoundError: [Errno 2] No such file or directory: '/some/path/to/miniforge3/envs/cxr_qba/lib/python3.11/site-packages/cxr_qba_extract/reference_data/regions.csv'
```

If you check `cxr_qba_extract/MANIFEST.in`, the path `include src/cxr_qba_extract/reference_data/**/*` is incorrect; it should be `include src/cxr_qba_extract/reference_data/*` and then `pip install ./cxr_qba_extract` will import the correct folder.

However, when I ran the `cxr_qba_extract status`, I get many NaN values which is confusing. I am not sure how to hotfix this. Please note that all paths have already been exported, so the sample count is correct. It is just that I got these values as NaN.
```
Left version is missing for regions with laterality:
region
lung lower lobe right     NaN
lung middle lobe right    NaN
lung upper lobe right     NaN
azygos lobe               NaN
...
```


**Import error**
```bash
cxr_qba_extract cxas
```

To run the above command, need to go to file `scripts/run_cxas.py` and fix:
```python
import sys
from ..cxas_mimic.extract_features import extract_features_for_folder
```
to correct the import packages

**PyTorch compatibility issue**
L90, file `/some/path/to/miniforge3/envs/cxr_qba/lib/python3.11/site-packages/cxas/models/__init__.py`

Change to:
```python
checkpoint = torch.load(out_path, map_location=map_location, weights_only=False)
```
This depends on PyTorch version. The newest one will require us to explicitly set `weights_only=False`.

**GPU checking logic**
L85 file `cxas_mimic/extract_features.py`

If `model.gpus = cuda:0`, it still passes the if condition. I understand it's checking list of cuda devices, but the default code doesn't seem to expect the value to be a string.

Based on this evidence, I cannot mark "Yes" for code accessibility as the current version is not in a full and final form. I hope the authors further check the code to be sure before publishing. However, this only affects a little to my rating as I appreciate the authors' work on making this code very rich in functionality.

**Ethical Comments:**

This work is based on another existing dataset (MIMIC-CXR), which is already de-identified and suitable for research purposes and satisfies many ethical requirements. And there is also no new patient data collected. So, there are no major ethical concerns that I am aware of.

**Ethical Considerations:**

No, there are no or only very minor ethics concerns

**Final Justification:**

My major concern about "a solely automatically generated dataset" is mostly addressed by the authors' confirmation of QA by an expert listed among the authors. However, I would like the authors to add these details in Section 3.3, as this would improve researchers' trust in the quality more than a simply automatic system would. I believe this would be a good contribution to the medical domain, as fine-grained large chest X-ray data is still a gap in the field that needs to be filled. Even if this is not a clinical-grade evaluation dataset, it is still a valuable data source for training, as it is better than Chest ImaGenome (Section 4.1). Besides, most of my other concerns are addressed as well.
In my scoring ladder, I raise my "Significance" score to 3 and have updated my decision score to 5.

**Limitations Weaknesses:**

### Quality
- Heavy reliance on LLM-based quality assessment without human validation undermines technical rigor. Section 4.2 feels particularly weak. A human user study on a smaller random subset would improve credibility than solely LLM-as-judge
- Limited evaluation of different LVLM models (only Llama 3.1 8B and 70B presented). Missing comparisons with other state-of-the-art models like GPT-4o, Qwen-VL, etc., to determine if better models could improve dataset quality
- The use case for "pathology localization" training (Section 6.1) appears inaccurate since bounding boxes are only provided for anatomy, not for findings or pathology

### Clarity
- Unclear explanations: What exactly is the positive ratio in Figure 5? Why is there a "not rated" category and what insights does it provide?
- Multiple formatting issues throughout (L803-804, 821-822, 861-862, L133 extra bracket, L970 tense inconsistency)
- Template numbers (13, 6, and 7) at L156-166 appear arbitrary without clear reasoning
- Steps L961-964 would benefit from illustrations as the text description is difficult to visualize
- Why isn't the "Hints" section combined with "Rules" in LLM prompts?

### Significance
- The authors acknowledge the dataset is not suitable as "a sole source" for "evaluating models used in clinical practice" (Section 6.2) due to LLM-based labels, which significantly limits its clinical applicability
- For medical domain applications, many readers would expect human-labeled data to establish actual benchmark value. The lack of a selected human-verified gold-standard subset limits practical utility

### Originality
- While the scale is novel, the core approach of using LVLMs for dataset generation is not fundamentally new
- Limited exploration of alternative LVLM architectures reduces the novelty of the technical approach

**Strengths Contributions:**

### Quality
- The work presents a technically sound automated pipeline that enables good scalability for dataset creation
- Claims are well-supported by comprehensive analysis showing that scene graphs contain plausible finding tags and bounding boxes with competitive or better quality than Chest ImaGenome (L208-209)
- The rationale for using Llama 3.1 8B instead of the 70B version is convincing and well-justified
- The comprehensive appendix with all prompt listings demonstrates methodological transparency
- The authors are honest about the paper's limitation.

### Clarity
- Well-organized structure with comprehensive appendix
- Clear presentation of the dataset creation pipeline
- Helpful inclusion of all prompt listings

### Significance
- Provides 42M QA-pairs with multi-granular, multi-part answers, bounding boxes, and structured tags - the largest and most sophisticated VQA dataset for CXRs to date
- Defines a significantly larger set of localized regions (257) and findings (221) compared to Chest ImaGenome (29 regions, 53 findings), making it substantially richer for VQA tasks
- Addresses scalability challenges in medical VQA dataset creation

### Originality
- Novel approach to automated VQA dataset generation for medical imaging at unprecedented scale
- Innovative combination of scene graph generation with multi-granular question-answer pair creation
- The proposed task of Structured VQA is also novel and interesting.

---

> ### Author Rebuttal · Authors · 2025-07-30
>
> ## **TL;DR**
> - We now conducted a systematic human assessment of the LLM-as-a-judge.
> - Preliminary experiments in late 2024 indicated that the used LLM provides one of the best trade-offs between quality and efficiency. Systematic comparisons of extraction models would be very expensive.
> - The pathology localization use case is possible as our dataset provides coarse bounding boxes for pathologies based on associated anatomical regions.
> - The dataset construction pipeline was co-designed with trained radiologists (from the author list) who performed QA at every step. In Sec. 4.1, we also validate the scene graphs against expert annotations. However, our dataset targets large-scale training while it is not intended for clinical-grade fine-tuning and evaluation, therefore we provide no gold-standard.
> - Our work focuses on proposing a dataset construction pipeline and contributing a dataset, while the task proposed in Sec. 5 is mainly meant as a demonstration of utility. Thus, we consider detailed studies on model-related design decisions out-of-scope.
>
> ## **Reliance on LLM-based quality assessment**
>
> We highlight that the dataset construction pipeline, including the LLM-as-a-judge, was co-designed with trained radiologists (from the author list) who performed QA at every step.
>
> Additionally, we have now **conducted a systematic human user study on 100 randomly sampled** cases (without replacement) for **each** of the LLM-as-a-judge’s five criteria (entailment, relevance, completeness, question clarity, answer clarity). Each of the sampled cases was rated by a human annotator provided with the same information available to the LLM-as-a-judge.
>
> Comparing the human ratings with the LLM ratings, we found that in a **maximum of 2%** of the cases the LLM judgment **lead to an overrated grade** (e.g. fine-tuning instead of pre-training grade) compared to human judgment. Overall, we found **high agreement** (using Cohen’s kappa, a standard metric for inter-rater reliability) with a tendency of the LLM to be more critical than human ratings (especially for completeness and clarity).
>
> Results are shown below:
>
> |  Metric                    | Entailment | Relevance | Completeness | Question clarity | Answer clarity |
> | :--------------------  | -----------: |-------------:|-----------------:|-------------------:|-----------------:|
> |                                | (7 classes) | (4 classes) | (5 classes)       | (6 classes)         | (5 classes)       |
> | % overrated grade | 1%            | 0%            | 2%                  | 0%                     | 0%                  |
> | Cohen's kappa       | 0.49          | 0.65           | 0.41                 | 0.32                   | 0.32                 |
>
> We will include these results and the detailed confusion matrices in the appendix and summarize them in Sec 4.2.
>
> Also note that the LLM-as-a-judge is only one part of the evaluation. **In Sec. 4.1 we also validate the scene graphs against expert annotations.**
>
> **(See also rebuttal to R. Qj4q)**
>
> ## **Limited evaluation of different LVLM models**
>
> We choose the information extraction model (Llama 3.1 70B) based on the following criteria:
> - Open weights and locally hostable (required due to the MIMIC dataset licenses, excludes models such as GPT-4o and Gemini)
> - Fast and efficient to run on large amounts of data
> - Sufficient extraction quality and instruction following
> - No vision component is required (i.e. we only need LLMs but not LVLMs)
>
> While using other models would be possible, we choose Llama 3.1 70B because preliminary experiments (at the start of dataset construction in late 2024) indicated that it **provides one of the best trade-offs of above’s points**.
>
> While the study of alternative models (like Qwen) would be interesting, these would require immense additional resources with little expected benefits, especially as our work does not intend to benchmark LLMs but instead use them as a tool for construction. However, we **release the full sourcecode for dataset construction, such that other models could be studied** in future work.
>
> Additionally, note that in **Appendix C.1 we studied the impact of the model size on the LLM-as-a-judge** quality assessments.
>
> ## **Use case for "pathology localization" training**
>
> While the bounding boxes are derived from anatomical regions, we associate them with pathologies based on what is written in the reports (i.e. mentions of where each pathology is located) and based on predefined knowledge (Sec 3.1 and Appendix E.1). While they may not always be perfectly accurate, they **provide a course localization for pathologies**. We will clarify this in Sec. 6.1.
>
> ## **Not suitable as "a sole source" for "evaluating models used in clinical practice"**
>
> We agree that this is a limitation but note that our dataset **targets pre-training and large-scale fine-tuning**, while it is not intended for clinical-grade fine-tuning and evaluation. For such use cases we recommend other small-scale but human-annotated datasets. We will further clarify this in Sec. 6.1.
>
> ## **Lack of a selected human-verified gold-standard**
>
> While we do not provide a human-verified gold-standard, **in Sec 4.1 we validate a subset of our constructed scene graphs** (finding bounding boxes and finding identification) against datasets manually annotated by medical experts (radiologist annotations in MIMIC-CXR-JPG,  CXR-LT 202, REFLACX, MS-CXR).
> While the lack of a gold-standard may limit its use as a clinical-grade benchmark, our contribution provides a large scale resource for training and prelimenary evaluation/benchmarking.
>
> ## **Using LVLMs for dataset generation is not fundamentally new**
>
> Our construction pipeline combines different existing techniques like LLM-based information extraction, localization, semantic entity linking, predefined knowledge definitions, as well as template- and rule-based QA generation. While individual techniques may not be fundamentally new, we argue that combining them to a reliable construction pipeline is a significant novel contribution. Also note that our approach does not rely on vision capabilities of LVLMs but uses LLMs instead.
>
> ## **Limited exploration of alternative LVLM architectures reduces the novelty of the technical approach**
>
> **Consistently with the D&B track**, our work focuses on proposing a dataset construction pipeline and contributing a dataset. Therefore, the task presented in Sec. 5 is meant as a demonstration of the utility of our dataset and to serve as a baseline for future work. Thus, we consider detailed studies on model-related design decisions (e.g. architectures) out-of-scope. However, **in Appendix F.1 (p. 43) we provide some ablation studies** on the impact of different ways to utilize our annotations.
>
> **(See also rebuttal to R. LYLZ)**
>
> ## **Clarity**
>
> We will address the mentioned clarity issues in the camera-ready version:
>
> - Positive ratio in Figure 5: as mentioned in the caption, this is the ratio of the number of mentions in a positive answers (i.e. tagged as "positive") versus the mentions in a negative answer.
> - “Not-rated” (e.g. in Fig. 4) occurs if the answers of the LLM-as-a-judge where not parsable.
> - Templates were defined in collaboration with trained radiologists (from the author list) and inspired by question types in other CXR VQA datasets. Thus, the number of templates is based on what we identified as relevant questions.
> - Prompt: We found adding additional “Hints” in the prompt after the examples to be beneficial.
> - We will fix the formatting issues for the camera-ready deadline.
>
> ## **Dataset Code**
>
> We will address the mentioned issues and re-test the codebase until the camera-ready deadline.

---

> > ### Comment · Reviewer_5Yi9 · 2025-08-04
> >
> > I thank the authors for their response. My major concern about "a solely automatically generated dataset" is mostly addressed by the authors' confirmation of QA by an expert listed among the authors. However, I would like the authors to add these details in Section 3.3, as this would improve researchers' trust in the quality more than a simply automatic system would. I believe this would be a good contribution to the medical domain, as fine-grained large chest X-ray data is still a gap in the field that needs to be filled. In my scoring ladder, I raise my "Significance" score to 3 and have updated my decision score to 5.

---

> > > ### Author Response · Authors · 2025-08-05
> > >
> > > Thank you for the productive discussion!
> > >
> > > We will include the suggested clarifications in Sec. 3.3 of the manuscript.

---

### Official Review · Reviewer_LYLZ · 2025-07-01

**Rating:** 4
**Confidence:** 3

**Summary:**

This paper introduces CXR-QBA, a large-scale visual question answering (VQA) dataset for chest X-ray (CXR) interpretation, automatically derived from the MIMIC-CXR database. The dataset comprises over 42M question-answer (QA) pairs, along with bounding box annotations, structured tags, and full-sentence, radiology-style answers. The authors propose an automated pipeline that constructs scene graphs from radiology reports and CXR images, generates QA pairs from these graphs, and conducts LLM-based quality assessments to filter for fine-tuning and pre-training quality data. The authors also define a new structured VQA task where the model should generate free-text answers accompanied by bounding boxes and tags (e.g., findings, regions), and introduce RadStrucVQA, an evaluation metric for this structured and grounded answers.

**Dataset Code Accessibility:**

Yes

**Dataset Code Comments:**

The dataset is fully accessible, and the benchmark code is well-documented to reproduce the experiments.

**Ethical Considerations:**

No, there are no or only very minor ethics concerns

**Final Justification:**

I've raised a concern regarding the model evaluation that has been performed with only one baseline and architecture.
However, after I read the authors' response and other reviewers' comments, I understand that the main contribution of the work lies in its dataset construction process and the dataset itself, and this concern is only minor for this submission.

**Limitations Weaknesses:**

* While the authors include a structured VQA task and introduce RadStrucVQA metrics, the evaluation is limited to a single model and one baseline (MAIRA-2), which is a critical limitation of this paper.

**Strengths Contributions:**

* The paper is well-written and organized.
* The proposed dataset (CXR-QBA) includes bounding boxes and structured tags for answers, enabling visually grounded and explainable VQA tasks, which makes the dataset unique and novel.
* The authors provide a fully automated pipeline to construct the dataset from building scene graphs using radiology reports and CXR images to generate and refine QA pairs using LLMs, which is a key contribution of the paper.

---

> ### Author Rebuttal · Authors · 2025-07-30
>
> ## **TL;DR**
> - Our work focuses on proposing a dataset construction pipeline and contributing a dataset, while the task proposed in Sec. 5 is mainly meant as a demonstration of utility. Thus, we consider detailed studies on model-related design decisions out-of-scope.
> - Note that no other baselines that are capable of solving the task in Sec. 5 are available.
>
> ## **Structured VQA task evaluation limited to single model and baseline**
>
> We point out that the task presented in Sec. 5 is mainly meant as a demonstration of the utility of our dataset, i.e.:
>
> - Showcase an example application, inspiring usage patterns of the dataset that were not possible with previous datasets.
> - Propose a simple method to serve as a baseline while leaving room for more advanced methods in future research.
>
> ### **Why only a single model?**
>
> **Consistently with the D&B track**, the focus of this work is to propose a dataset construction pipeline and to contribute a dataset.
> The task presented in Sec. 5 is mainly meant as a demonstration of the utility of our dataset and to serve as a baseline, whereas we consider detailed studies on model-related design decisions (such as architecture, model size) out-of-scope. However, in Appendix F.1 (p. 43) we **provide some ablation studies** on the impact of different ways to utilize our annotations.
>
> We also note that the baseline MAIRA-2, a different model trained on the original reports, performs quite good in “logical recall”, i.e. predicts most of the expected answers, confirming that adjusting existing models can also be a strategy to solve the task (see Sec. 5).
>
> ### **Why only one baseline?**
>
> As the structured VQA task has not been tackled before, **there are no baselines specifically targeting this task**. The used **MAIRA-2 baseline** is the most relevant, and to the best of our knowledge, **only publicly available chest X-ray report generation (or VQA) model that supports predicting bounding boxes** and that is therefore at least partially able to solve the structured VQA task. We included it to (i) check the plausibility of our task and annotations, and (ii) to see how much benefit the use of our VQA samples provides compared to sole report generation training.
>
> Again, **consistently with the D&B track**, we focus on dataset-related aspects and consider detailed studies and comparisons of model-related aspects out-of-scope.

---

> > ### Comment · Reviewer_LYLZ · 2025-08-02
> >
> > Thank you for the response. I have left my comments below.
> >
> > I understand that the core contribution of this work lies in the dataset construction pipeline and the dataset itself. Although I still believe that, as a dataset provider, the authors should also provide the potential utility of the dataset, including the expected tasks that can be performed with this dataset and the reference performances in general, this is only minor when considering the main contribution of the work.
> >
> > Therefore, considering the quality of the work in general, I would raise my score up to 4.

---

> > > ### Author Response · Authors · 2025-08-05
> > >
> > > Thank you for your constructive feedback.
> > >
> > > We will include a discussion of your point ("potential utility of the dataset") in Sec. 6 of our manuscript.

---

### Official Review · Reviewer_Uohr · 2025-07-04

**Rating:** 4
**Confidence:** 4

**Summary:**

This paper introduces CXR-QBA, a large-scale dataset for Visual Question Answering (VQA) on chest X-rays, derived from MIMIC-CXR. The authors develop an automatic pipeline that creates 42.2M QA-pairs with multi-granular text answers accompanied by bounding boxes and tags. The dataset is evaluated with automatic quality assessment, and also a proof-of-concept structured VQA model.

**Dataset Code Accessibility:**

Yes

**Dataset Code Comments:**

The authors released the dataset on a public web link, containing detailed instructions and python files for the dataset.

**Ethical Considerations:**

No, there are no or only very minor ethics concerns

**Final Justification:**

Thanks for the response. The newly included systematic human user study is helpful to evaluate the quality of the constructed dataset. But, the answer to the 3rd question somehow degrades the importance of the proposed dataset, which claimed that the dataset targets pre-training and fine-tuning, not for clinical-grad fine-tuning and evaluation. Hence, I change my final score to 4.

**Limitations Weaknesses:**

1. The quality assessment relies heavily on LLM judges (Llama 3.1 8B), which may introduce systematic biases. Conducting human validation on a substantial subset is needed to calibrate LLM judges and establish inter-rater reliability metrics.
2. The model evaluation is limited to a single architecture, which makes it difficult to assess the dataset's general utility.
3. In the validation part, the paper lacks validation from radiologists or clinical experts. Given the clinical nature of the data, expert validation is crucial for establishing medical accuracy.

**Strengths Contributions:**

1. The paper addresses a critical need in medical AI by providing the first large-scale structured VQA dataset for chest X-rays.
2. The automatic three-stage pipeline to construct the dataset is novel and technically sound, which involves scene graph construction, QA generation, and automatic quality assessment.
3. The dataset has a high quality compared to existing medical report generation dataset (see Table 2). It provides localization annotations and structured metadata, covering 221 finding classes and 257 region classes.
4. The paper is well-structured with clear methodology descriptions, informative figures and comprehensive appendices.

---

> ### Author Rebuttal · Authors · 2025-07-30
>
> ## **TL;DR**
> - We now conducted a systematic human assessment of the LLM-as-a-judge.
> - Our work focuses on proposing a dataset construction pipeline and contributing a dataset, while the task proposed in Sec. 5 is mainly meant as a demonstration of the dataset's utility. Thus, we consider detailed studies on model-related design decisions out-of-scope.
> - The dataset construction pipeline was co-designed with trained radiologists (from the author list) who performed QA at every step. In Sec. 4.1, we also validate the scene graphs against expert annotations.  However, our dataset targets large-scale training while it is not intended for clinical-grade fine-tuning and evaluation (see. Sec. 6.2)
>
> ## **(1) Quality assessment relies heavily on LLM judges**
>
> We highlight that the dataset construction pipeline, including the LLM-as-a-judge, was co-designed with trained radiologists (from the author list) who performed QA at every step.
>
> Additionally, we have now **conducted a systematic human user study on 100 randomly sampled** cases (without replacement) for **each** of the LLM-as-a-judge’s five criteria (entailment, relevance, completeness, question clarity, answer clarity). Each of the sampled cases was rated by a human annotator provided with the same information available to the LLM-as-a-judge.
>
> Comparing the human ratings with the LLM ratings, we found that in a **maximum of 2%** of the cases the LLM judgment **lead to an overrated grade** (e.g. fine-tuning instead of pre-training grade) compared to human judgment. Overall, we found **high agreement** (using Cohen’s kappa, a standard metric for inter-rater reliability) with a tendency of the LLM to be more critical than human ratings (especially for completeness and clarity).
> Results are shown below:
>
> |  Metric                    | Entailment | Relevance | Completeness | Question clarity | Answer clarity |
> | :--------------------  | -----------: |-------------:|-----------------:|-------------------:|-----------------:|
> |                                | (7 classes) | (4 classes) | (5 classes)       | (6 classes)         | (5 classes)       |
> | % overrated grade | 1%            | 0%            | 2%                  | 0%                     | 0%                  |
> | Cohen's kappa       | 0.49          | 0.65           | 0.41                 | 0.32                   | 0.32                 |
>
> We will include these results and the detailed confusion matrices in the appendix and summarize them in Sec 4.2.
>
> Note that a systematic human evaluation on a much larger subset would be very expensive and time-consuming. However, we **provide additional quality assessments in Sec 4.1**, where we compare our scene graphs (their finding bounding boxes and finding identification) against datasets manually annotated by medical experts (radiologist annotations in MIMIC-CXR-JPG,  CXR-LT 202, REFLACX, MS-CXR)
>
> **(See also rebuttal to R. Qj4q)**
>
> ## **(2) Model evaluation is limited to a single architecture**
>
> **Consistently with the D&B track**, the focus of this work is to propose a dataset construction pipeline and to contribute a dataset. The task presented in Sec. 5 is mainly meant as a demonstration of the utility of our dataset and to serve as a baseline for future work. Thus, we consider detailed studies on model-related design decisions (e.g. architectures) out-of-scope and leave it to future work. However, in Appendix F.1 (p. 43) we provide some ablation studies on the impact of different ways to utilize our annotations.
>
> We also note that MAIRA-2, a different model trained on the original reports, performs quite good in “logical recall”, i.e. predicts most of the expected answers, confirming that adjusting existing models can also be a strategy to solve the task (see Sec. 5).
>
> **(See also rebuttal to R. LYLZ)**
>
> ## **(3) Lack of validation from radiologists or clinical experts**
>
> While we did not conduct a systematic validation with radiologists or clinical experts, we highlight that **trained radiologists are on the author list and co-designed the dataset construction pipeline**, including (i) designing reference definitions for entity mapping, (ii) checking extracted scene graphs and failure cases, and (iii) analysis of exemplary generated QA pairs, including the ones shown in Appendix A.
>
> We also highlight that **in Sec 4.1 we validate our constructed scene graphs** (finding bounding boxes and finding identification) against datasets manually annotated by medical experts (radiologist annotations in MIMIC-CXR-JPG,  CXR-LT 202, REFLACX, MS-CXR).
>
> Finally, in Sec. 6.2 we “strictly advise against using this dataset as the sole source for fine-tuning or evaluating models used in clinical practice”. Our dataset targets pre-training and large-scale fine-tuning, while it is **not intended for clinical-grade** fine-tuning and evaluation, for which we recommend other small-scale but human-annotated datasets.

---

### Official Review · Reviewer_Qj4q · 2025-07-19

**Rating:** 4
**Confidence:** 5

**Summary:**

This paper proposes a new CXR VQA dataset, namely CXR-QBA,  comprising 42 million QA-pairs with multi-granular, multi-part
8 answers, detailed bounding boxes, and structured tags. Based on data quality, the authors also split into two subset, i.e.,  31M for pre-training and 7.5M  for fine-tuning. Generally, I think this paper is good, but it still has some shortcomings that should be polished.

**Dataset Code Accessibility:**

Yes

**Dataset Code Comments:**

The data is clearly released.

**Ethical Considerations:**

No, there are no or only very minor ethics concerns

**Final Justification:**

Thank you very much for the revision. Most of my concerns have been addressed, particularly regarding the differences compared to Chest ImaGenome and the reliance on LLM-as-judge. I have updated my final score to 4.

However, since current medical VQA datasets are predominantly based on CXR (which is why I was curious about a clear explanation of the differences from Chest ImaGenome), this somewhat limits the overall impact of the work. I believe we should encourage more research on other modalities, such as CT and MRI. Therefore, I cannot be more positive than 4.

**Limitations Weaknesses:**

Although the paper is generally well-executed, there are still several points that should be addressed:

1. In Section 3.3, the authors mention that LLM-as-a-judge is adopted; however, no metrics-based assessment is provided to reliably quantify its effectiveness.

2. The differences between this work and Chest ImaGenome should be elaborated on in greater detail, as this work seems to rely heavily on it from my perspective.

3. In Table 1, it should be explicitly stated that only 7.5M instances are of high quality for fine-tuning, while other datasets, such as VQA-RAD, are primarily intended for instruction tuning.

If these comments are adequately addressed, I would be very willing to raise my final score.

**Strengths Contributions:**

1. The dataset is substantial and highly meaningful.
2. The analysis of the dataset is comprehensive and well-executed.

---

> ### Author Rebuttal · Authors · 2025-07-30
>
> ## **TL;DR**
> - We now conducted a systematic human assessment of the LLM-as-a-judge.
> - We will highlight the differences to Chest ImaGenome. Note that our work does not rely on their scene graph structure but only their bounding boxes.
> - We will extend Tab. 1.
>
> ## **(1) LLM-as-a-judge Assessment**
> We highlight that the dataset construction pipeline, including the LLM-as-a-judge, was co-designed with trained radiologists (from the author list) who performed QA at every step.
>
> Additionally, we have now **conducted a systematic human study** to reliably quantify the effectiveness of the LLM-as-a-judge. For each of the five quality criteria (entailment, relevance, completeness, question clarity, answer clarity), we independently sampled 100 cases (without replacement). Each of these cases was rated by a human annotator provided with the same information available to the LLM-as-a-judge (see Appendix D.3, p. 26).
>
> Comparing the human ratings with the LLM ratings, we found that in a **maximum of 2%** of the cases the LLM judgment **lead to an overrated grade** (e.g. fine-tuning instead of pre-training grade) compared to human judgment. Overall, we found **high agreement** (using Cohen’s kappa, a standard metric for inter-rater reliability) with a tendency of the LLM to be more critical than human ratings (especially for completeness and clarity).
>
> Results are shown below:
>
> |  Metric                    | Entailment | Relevance | Completeness | Question clarity | Answer clarity |
> | :--------------------  | -----------: |-------------:|-----------------:|-------------------:|-----------------:|
> |                                | (7 classes) | (4 classes) | (5 classes)       | (6 classes)         | (5 classes)       |
> | % overrated grade | 1%            | 0%            | 2%                  | 0%                     | 0%                  |
> | Cohen's kappa       | 0.49          | 0.65           | 0.41                 | 0.32                   | 0.32                 |
>
> We will include these results and the detailed confusion matrices in the appendix and summarize them in Sec 4.2.
>
> Also note that the LLM-as-a-judge is only one part of the evaluation. **In Sec. 4.1 we also validate the scene graphs against expert annotations.**
>
> ## **(2) Differences to Chest ImaGenome**
>
> We point out that this work **does not rely on** the scene graph structure, sentences, or finding annotations from Chest ImaGenome. We only use their region bounding boxes as one source for our bounding boxes, while we rely on the CXAS segmentation model as the primary source for most regions. Our scene graphs are constructed based on the information in the original reports.
>
> **Comparison to the Chest ImaGenome dataset (partially provided in Sec. 2):**
>
> - Chest ImaGenome relies on rule-based information extraction, while we rely on LLM-based extraction with semantic entity mapping.
> - Chest ImaGenome uses only 29 regions and 53 findings, while we define 310 different regions (257 with bounding boxes) and 221 findings.
> - Unlike Chest ImaGenome, we rewrite report sentences to describe only specific aspects related to individual scene graph nodes.
> - We provide an additional scene graph node for the indication section and its related findings, which is not provided in Chest ImaGenome’s graphs.
> - Unlike Chest ImaGenome, we also generate question-answer pairs.
>
> We will extend/clarify Sec. 2 and 3.1 accordingly.
>
> ## **(3) Table 1 (number of fine-tuning instances)**
>
> We will add an additional row to Table 1 to explicitly highlight that only 7.5M of the instances from our dataset are of sufficiently high quality for fine-tuning.

---

> ### Comment · Reviewer_Qj4q · 2025-08-05
>
> Thank you for the clear explanation. As I promised, I'll raise my final score to 4 accordingly.
>
> However, since current medical VQA datasets are predominantly based on CXR (which is why I was curious about a clear explanation of the differences from Chest ImaGenome), this somewhat limits the overall impact of the work. I believe we should encourage more research on other modalities, such as CT and MRI. Therefore, I cannot be more positive than 4.

---

> > ### Author Response · Authors · 2025-08-05
> >
> > Thank you for your constructive feedback!
> >
> > We will extend the discussion of this limitation (focus only on CXR) in Sec. 6.2 of our manuscript.

---

### Comment · Area_Chair_zjnQ · 2025-08-04

Authors are anxious to know whether their responses have clarified your concerns, and whether there are any further questions or points that need addressing.

If you see something, say something — your input is greatly appreciated.

Many thanks to reviewers Uohr and LYLZ for promptly acknowledging the rebuttal.

---

### Note · Authors · 2025-08-12

We thank the reviewers and AC for a very constructive discussion phase.

We engaged with all feedback, providing additional analysis and clarifications that we feel have substantially improved the quality of the manuscript.

We are pleased that **all reviewers have converged on a positive opinion** for this paper, and we especially appreciate the discussions that helped us resolve major concerns, resulting in **increased scores from all three reviewers who participated in the discussion**.

---

### Decision · Program_Chairs · 2025-09-18

**Decision:**

Reject

**Comment:**

This paper presents a CXR VQA dataset generation pipeline that creates multi-granular question–answer pairs accompanied by bounding boxes and structured tags. The proposed three-stage pipeline builds on the methodology of Chest ImaGenome, but scales it up substantially, producing 42 million QA pairs covering 257 anatomical regions and 221 findings (versus 29 and 53 respectively in Chest ImaGenome). The reviewers broadly appreciated the unprecedented scale and the potential value of such a richly annotated dataset.

However, two concerns were raised:

Heavy reliance on LLM-based quality assessment, with limited human verification.

Limited methodological novelty, as the pipeline largely extends Chest ImaGenome without introducing fundamentally new techniques.

During the discussion, the authors provided a systematic human user study on 100 randomly sampled cases, which addresses part of the concern regarding annotation quality. This additional effort was positively acknowledged by the reviewers.

Nonetheless, the methodological incrementalism relative to Chest ImaGenome remains, and given the limited space for publication of NeurIPS, the decision is reluctantly to recommend rejection, despite recognising the value of the large-scale dataset and the additional validation effort.